# Effects of Flue Gas Impurities on the Performance of Rare Earth Denitration Catalysts

**Xue Bian \*, Kaikai Lv, Ming Cai, Peng Cen and Wenyuan Wu**

Key Laboratory of Ecological Metallurgy of Multi-Metal Intergrown Ores of Ministry of Education, School of Metallurgy, Northeastern University, Shenyang 110819, China; 2171640@stu.neu.edu.cn (K.L.); 1910573@stu.neu.edu.cn (M.C.); cenpeng@smm.neu.edu.cn (P.C.); wuwy@smm.neu.edu.cn (W.W.)
\* Correspondence: bianx@smm.neu.edu.cn

**Abstract:** Selective catalytic reduction (SCR) is still the most widely used process for controlling $NO_x$ gas pollution. Specifically, commercial vanadium-based catalysts have problems such as narrow operating temperature range and environmental pollution. Researchers have developed a series of cerium-based catalysts with good oxygen storage performance and excellent redox performance of $CeO_2$. However, the anti-poisoning performance of the catalyst is the key to its application. There are many kinds of impurities in the flue gas, which has a huge impact on the catalyst. The deposition of substances, the reduction of active sites, the reduction of specific surface area, and the reduction of chemically adsorbed oxygen will affect the denitration activity of the catalyst to varying degrees, and the poisoning mechanism of different impurities on the catalyst is also different. Therefore, this review divides the impurities contained in flue gas into different types such as alkali metals, alkaline earth metals, heavy metals, and non-metals, and summarizes the effects and deactivation mechanisms of various types of impurities on the activity of rare earth catalysts. Finally, we hope that this work can provide a valuable reference for the development and application of $NH_3$-SCR catalysts for rare earth denitration in the field of $NO_x$ control.

**Keywords:** rare earth catalyst; flue gas impurities; denitration activity; poisoning mechanism

## 1. Introduction

Nitrogen oxides ($NO_x$) cause a series of air pollution problems such as acid rain, photochemical smog, and ozone depletion [1]. Typically, $NO_x$ emissions are produced by stationary or mobile sources, including coal-fired power plants and automobile engines. In order to reduce the emission of $NO_x$, various denitration technologies such as SNCR (non-selective catalytic reduction), SCR (selective catalytic reduction), and SNCR-SCR combined method have been developed. Among them, the SCR method is a mature and efficient denitration method [2]. Selective catalytic reduction (SCR) with $NH_3$ is the best technology for $NO_x$ removal in terms of removal efficiency, stability and cost. At present, most commercial SCR denitration catalysts are vanadium-based catalysts, which are widely used due to their better thermal stability and denitration efficiency, but vanadium-based catalysts also have obvious shortcomings. For example, the working temperature range is relatively narrow (300~400 °C), and they exhibit poor resistance to alkali metals, heavy metals, sulfur dioxide, etc. However, because vanadium is highly toxic, usage of catalytic material with large amounts of vanadium can lead to further hazard waste environmental problem [3]. Therefore, the development of alternative low-temperature SCR catalysts has attracted much attention in recent years.

In order to solve the problems existing in vanadium-based catalysts, a series of "environmental friendly" catalysts were born; for example, Cu- and Fe-exchanged zeolite catalysts showed good SCR activities [4–7]. Other nontoxic transition metal oxide-based catalysts such as $CeO_2$-$TiO_2$-based catalysts [8,9], $FeO_x$-$TiO_2$-based catalysts [10–13], and

$CeO_2$-$WO_x$-based catalysts [14,15], etc. have also been investigated as potential alternatives. But rare earth-based catalysts stand out among these catalysts. Today, rare earth-based denitration catalysts have become a research focus. The rare earth elements represented by cerium oxide have excellent oxygen storage and redox properties and unique redox pairs $Ce^{3+}$/$Ce^{4+}$ [16–19], and was used as the active component of the catalyst, such as $CeO_2$-$TiO_2$ [20], $CeO_2$/$WO_3$ [21,22], $CeO_2$/$WO_3$-$TiO_2$ [23,24], $CeO_2$-$MoO_3$-$TiO_2$ [25,26], $Ce$/$TiZrO_x$ [27], $Ce_2$/$Cu_4Al_1O_x$ [28] and $Ce$-$Cu$/$TiO_2$ [29]. The results of numerous studies have shown that the rare earth-based catalysts have wider denitration temperature range, higher efficiency, and higher resistance to $SO_2$, especially the cerium-based polymetallic oxide catalysts, which further improve the redox performance, surface acidity, and resistance to $H_2O$/$SO_2$. For example, Mn-Ce/$TiO_2$ catalyst, $MnO_x$ have many variable valence states. Its oxides can be interconverted and exhibit good catalytic activity at low temperature [30–33]. Meanwhile, ceria can reduce the loss of specific surface area and pore volume during calcination, thereby improving the oxygen storage capacity and redox performance of the catalyst. Therefore, it has great potential in the field of low-temperature SCR catalysts [16,17,34–40], promising to replace traditional vanadium-based catalysts.

Rare earth-based flue gas denitration catalysts have broad application prospects in sintering, coking, cement, glass and other industries. Although its denitration performance is excellent, the denitration process performs well, there are still some problems to be solved for rare earth-based catalysts. The sulfur dioxide, alkali metals, alkaline earth metals, heavy metals and other impurities contained in the flue gas will irreversibly affect the denitration efficiency and service life of the catalyst [41]. In addition, there are a large number of non-metallic impurities in the flowing gas of coal-fired boilers and municipal solid waste incinerators, such as phosphorus, hydrogen halide, etc., and the influence of these impurities on the catalyst cannot be ignored. Therefore, researchers have carried out a lot of research on the influence and mechanism of different impurities on the denitration performance of catalysts. In this paper, the impurities in flue gas are classified, and the influence rules of different types of impurities on the performance of rare earth denitration catalysts are summarized, in order to provide a reference for the development and application of rare earth denitration catalysts.

## 2. The Influence of Metal Impurities

It is well known that the influence of metal impurities on the catalyst is very large, including alkali metals, alkaline earth metals and heavy metals. Among these metal impurities are K, Na, Pb, and their compounds. These substances deposited on the surface of the catalyst, thus reducing the SCR activity of the catalyst [42]. Next, the specific effects of these metal impurities on the catalyst are discussed.

### 2.1. Influence of Alkali Metals

The alkali metal elements are the most harmful element to chemically poison a catalytic material, including alkali metal oxides, alkali metal sulfates and alkali metal chlorides [43]. These substances typically come from fluid gases from static sources, such as coal, biomass, power plants, etc. Studies had shown that in high mobility gases, the hydrolysis or ion exchange of alkali metals (K and Na) will neutralize the acid sites on the catalyst surface, and the presence of alkali metals will also reduce the amount of $NH_3$ adsorbed by the catalyst. It has a serious deactivation effect on traditional vanadium-based catalysts, and the deactivation is more serious with the increase of its content. In addition, the different forms of alkali metals have different effects on catalyst activity. Many researchers have studied the influence of alkali metals on vanadium-based catalysts, but the research on cerium-based catalysts is not in-depth. With the promotion of cerium-based catalysts, it is necessary to study the influence of alkali metals on cerium-based catalysts from the perspective of industrial application.

Peng et al. [22]. studied the effect and mechanism of K and Na on the performance of $CeO_2$-$WO_3$ catalysts. Figure 1a showed the activity comparisons of V-W/Ti and CeW catalysts and corresponding 1 wt% K-doped catalysts at temperature ranging from 100 to 300 °C under a GHSV of 60,000 $h^{-1}$. Without K doping, the activity of the CeW catalyst was slightly higher than the V-W/Ti catalyst below 280 °C, with a maximum of nearly 99% $NO_x$ conversion at 220 °C, maintained up to 300 °C. During the $NO_x$ conversion test, when 1% K was loaded, the activity of the V-W/Ti catalyst dropped to 20% at 200 °C, while the CeW catalyst remained above 70% at the same temperature. It can be concluded that the CeW catalyst is more resistant to alkali metals than the traditional vanadium-based catalyst V-W/Ti below 300 °C. For CeW catalyst, The Na&K catalyst was less active at low temperatures but yielded higher $NO_x$ conversion above 200 °C compared with the 1 K and 0.58 Na catalysts. At a given molar concentration, K gave rise to more deactivation than Na below 200 °C, due to its more potent neutralizing properties (Figure 1b). At the same time, with the increase of the concentration of alkali metal loaded on the catalyst, the denitration efficiency of the catalyst decreases more obviously at the same temperature. $NH_3$-TPD, in-situ infrared (DRIFTS), $H_2$-TPR analysis, and DFT calculation showed that K and Na decreased the content of acid sites on the catalyst surface (Figure 2), thereby reducing the $NH_3$ gas adsorption amount and weakening the denitration performance. In addition, the DFT calculation in Figure 3 $H_2O$ can bond to the surface with a hydrogen atom interacting with the moved oxygen (1.38 Å). The bond length of H–O in $H_2O$ was 1.11 Å, which was longer than the standard H–O bond in $H_2O$ (0.98 Å) and the other H–O bond is remained unchanged. This model indicated that the hydrolysis dissociation is $H^+$ and $OH^-$, and $H^+$ combines with the O element on the catalyst surface to form the Brønsted acidic site. Therefore, in the experiment, the denitration efficiency of the poisoned catalyst can be restored to 90% of that of the fresh catalyst after hot water cleaning (Figure 1c). Du et al. [3]. also used DFT calculation to find that the strong interaction of K ions with cerium oxide and titanium oxide reduced the oxygen vacancy and $NH_3$ gas adsorption of the catalyst, thereby reducing the reduction and denitration performance of the cerium-titanium oxide catalyst.

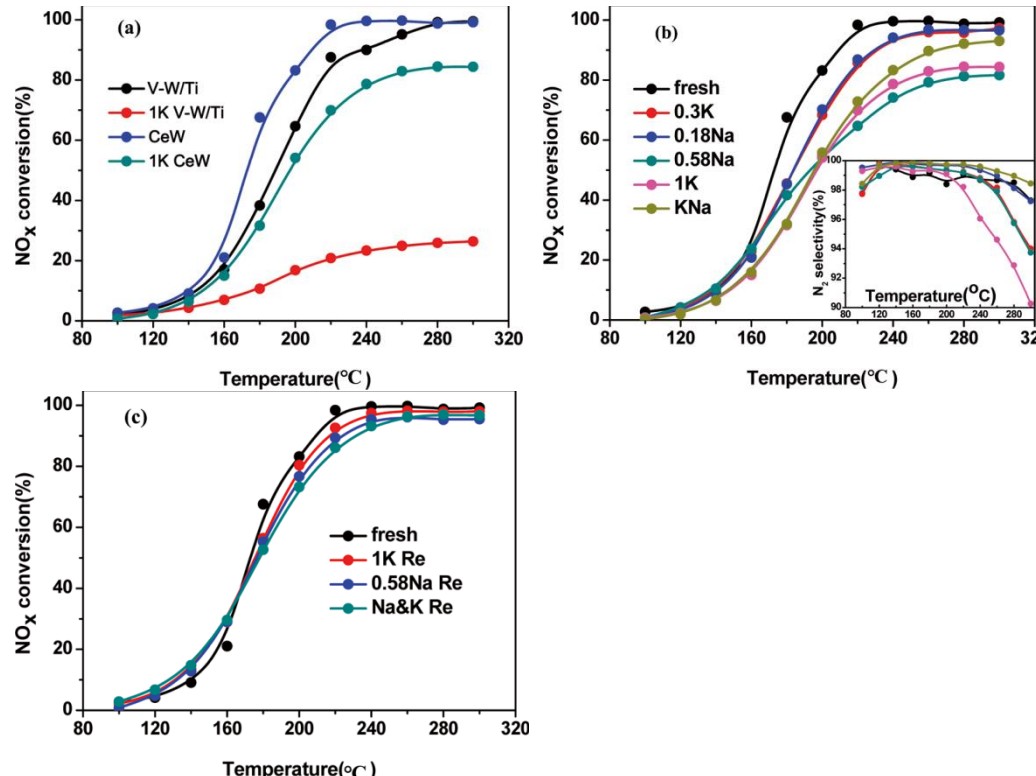

**Figure 1.** (**a**) Comparison of NH₃−SCR activity of V−W/Ti and CeW catalysts with corresponding 1 wt% K-doped catalysts; (**b**) The NH₃−SCR activity and N₂ selectivity of CeW and alkali-doped Ce W catalysts; (**c**) The NH₃−SCR activity of the regenerated poisoned catalysts. Reaction conditions: catalyst = 300 mg, [NO] = [NH₃] = 500 ppm, [O₂] = 3%, total flow rate = 300 mL/min, GHSV = 60,000 h⁻¹ [22].

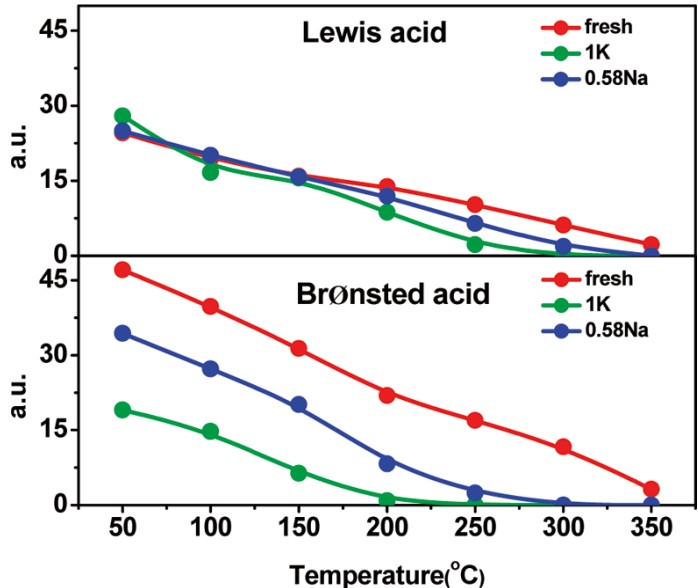

**Figure 2.** The strength and quantity of Lewis and Brønsted acid sites calculated from DRIFTS spectra, centered at 1160 cm⁻¹ (Lewis acid) and 1430 cm⁻¹ (Brønsted acid), respectively [22].

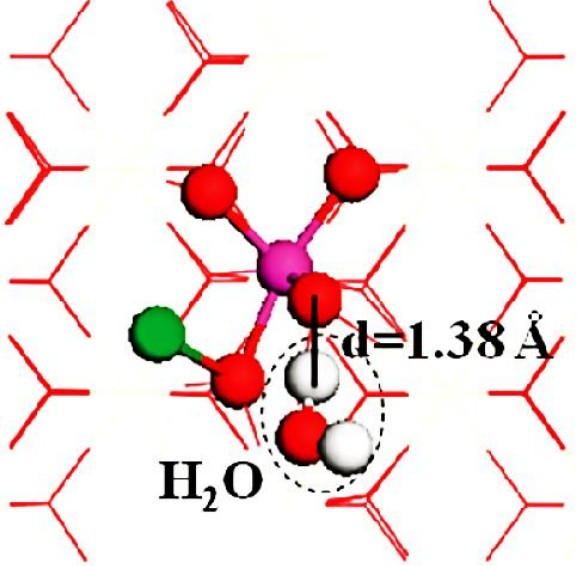

**Figure 3.** H₂O adsorbed on the K-poisoned (110) catalyst surface [22].

Wang et al. [44]. studied the effect of alkali metal K and Na loaded on the denitration performance of Ce/TiO₂ catalyst by coprecipitation method. The results showed that the denitration efficiency of fresh catalyst reached 90% at 200 °C, while the denitration efficiency of Na and K loaded with 0.2 mol ratio of Ce to Na decreased to 25% and 10% respectively at this temperature. Zhou's study [45] showed that the influence of alkali metal salts on Ce-Ti catalyst was in the order of nitrate < chloride < carbonate. The catalyst was

almost completely inactivated when potassium carbonate was loaded. When sodium carbonate was loaded, the denitration efficiency at 100 °C decreased from more than 90% to about 15%.

In addition, Jiang et al. [46]. also studied the influence of alkali metal compounds sodium oxide and sodium chloride on the denitration performance of $CeO_2$-$TiO_2$ catalyst. The results in Figure 4 showed that both sodium oxide and sodium chloride lead to catalyst deactivation, and the effect of sodium oxide is greater than that of sodium chloride. At 350 °C, when the molar ratio of Na: Ce = 0.5 $Na_2O$ was added, the catalyst almost had no denitration effect. When NaCl with molar ratio of Na: Ce = 0.5 was added, the denitration performance of the catalyst decreased to 60%. The XRD patterns of the fresh and Na-poisoned CT samples indicated the interaction between Na species and $TiO_2$ (Figure 5a). From the results of XPS, $NH_3$-TPD (Figure 5b) and in situ DRIFT studies, it was found that the addition of Na could inhibit the transformation of $Ce^{4+}$ to $Ce^{3+}$, sodium oxide, and sodium chloride decreased the specific surface area, chemical adsorption of oxygen, surface acidity, and reduction capacity of the catalyst, inhibited the adsorption of $NH_3$, and reduced the denitration performance. The denitration reaction is still controlled by E-R and L-H mixed mechanism when the catalyst is loaded with sodium oxide and sodium chloride.

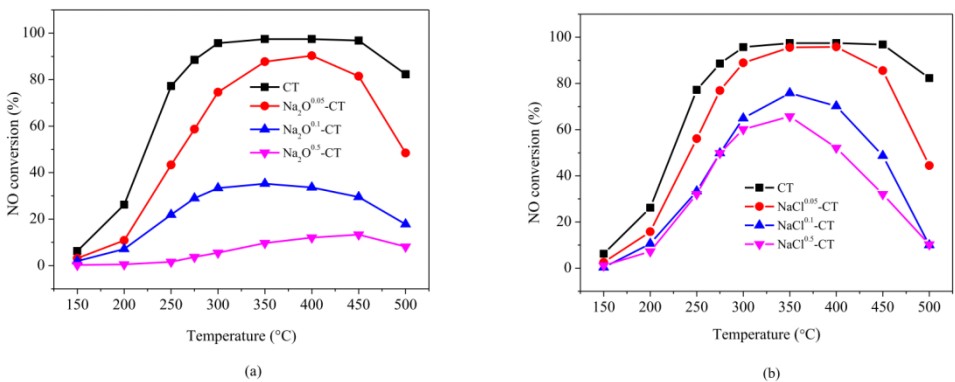

**Figure 4.** NO conversion of $Na_2O^{0.5}$-CT (**a**), $NaCl^{0.5}$-CT (**b**). Reaction condition: [NO] = [$NH_3$] = 1000 ppm, [$O_2$] = 3 vol%, $N_2$ balance, total flowrate = 500 mL/min, GHSV = 90,000 h$^{-1}$ [46].

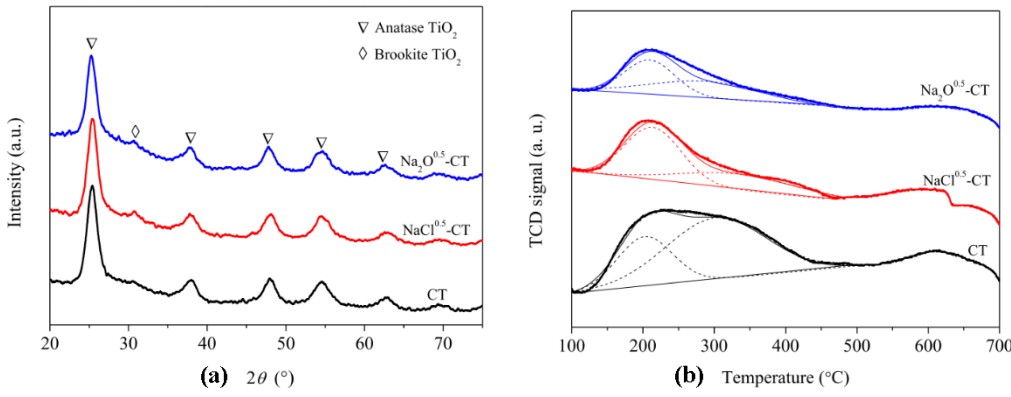

**Figure 5.** (**a**) XRD patterns of CT, $Na_2O^{0.5}$-CT, and $NaCl^{0.5}$-CT catalysts; (**b**) $NH_3$-TPD profiles of CT, $Na_2O^{0.5}$-CT, and $NaCl^{0.5}$-CT catalysts [46].

Similarly, for $CeO_2$-$TiO_2$ catalyst, the poisoning effect of $K_2O$ is more serious than that of KCl, as shown in Figure 6. When $K_2O$ with molar ratio K: Ce = 0.5 was added below 300 °C, the catalyst was almost completely deactivated. The characterization results showed that, compared to KCl, $K_2O$ could significantly reduce the surface acidity, reduction,

$Ce^{3+}/Ce^{4+}$ ratio, and the concentration of surface chemisorption oxygen of $CeO_2$-$TiO_2$ catalyst. In-situ DRIFT results showed that $K_2O$ had a stronger inhibitory effect on $NH_3$ adsorption on the catalyst surface than KCl. The introduction of $K_2O$ or KCl promoted the adsorption of NO on the catalyst surface, but not all $NO_x$ species were reactive in $NH_3$-SCR reaction. These results are further confirmed by DFT calculations [47]. The PDOS, oxygen vacancy, chemical oxygen adsorption, $NH_3$ and $NO_x$ adsorption energies of $CeO_2$-$TiO_2$ catalysts doped with different K species were calculated by MS Dmol 3. It was found that the introduction of K species weakened the reaction activity on the catalyst surface, inhibited the formation of oxygen vacancies and chemical adsorbed oxygen, and reduced the adsorption of $NH_3$ on the catalyst surface, all of which led to the decrease of catalytic activity [48].

The above studies show that the existence forms and types of alkali metals have different effects on cerium-based catalysts.

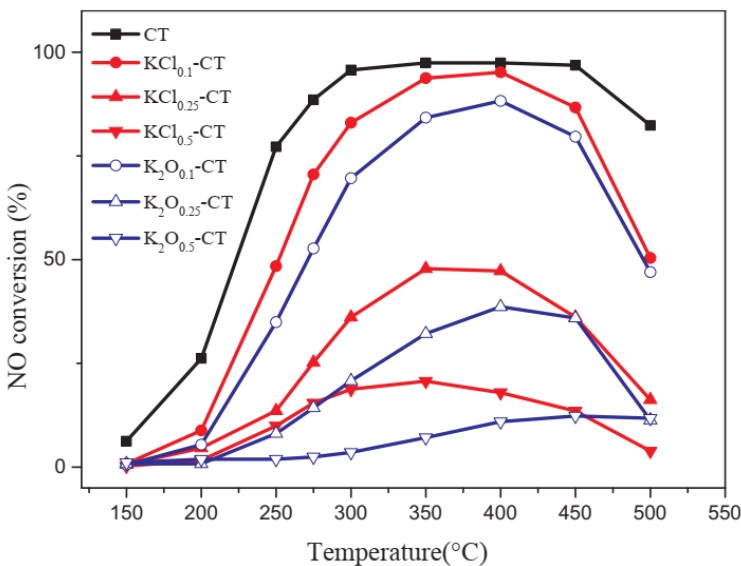

**Figure 6.** NO conversion of $KCl_x$-CT, $K_2O_x$-CT and CT. Reaction condition: [NO] = [$NH_3$] = 1000 ppm, [$O_2$] = 3%, $N_2$ balance, total flow rate = 500 mL/min, GHSV = 90,000 $h^{-1}$ [48].

## 2.2. Effect of Alkaline Earth Metal Ca

Although the low temperature SCR denitration reactor is mostly arranged after the dust removal system, there is still a small amount of residual dust in the flue gas. The catalyst is exposed to the flue gas with complex components for a long time, and it is easy to be inactivated by K, Na, Ca, Si, and As [49], especially when the low temperature SCR denitration technology is used in some industrial furnaces, such as glass furnaces, cement furnaces, etc. A large amount of alkaline earth metal Ca in flue gas will have a great influence on the activity of the catalyst [50,51]. The common way of catalyst poisoning caused by alkaline earth metal is that alkaline earth metal oxides (such as CaO) react with $SO_3$ in their pores to form calcium sulfate, which causes the pores to be blocked. The results of XRD on the catalyst surface by Benson et al. [52] showed that the alkaline earth metal compounds deposited on the catalyst surface are mainly $CaSO_4$, and the rest are $Ca_3Mg$ $(SiO_4)_2$ and $CaCO_3$. Among them, $CaSO_4$ and $CaCO_3$ are obtained by the reaction of CaO with $SO_3$ and $CO_2$, respectively. In addition, similar to alkali metals, alkali-earth metals can also interact with the Brønsted acid sites on the catalyst surface to cause the chemical poisoning of the catalyst, but due to the weak alkaline limit, the poisoning effect is relatively small [53].

The doping of Ca into the catalyst will have a certain effect on the structure, acid site and activity of the catalyst. Liu et al. [54] doped Ca element in $MnO_x/TiO_2$ catalyst found

that Ca doping could have negative effect on the activities of the $MnO_x/TiO_2$ catalysts. However, at a high Ca doping level of 10 wt%, this effect would become dilute, the SCR activity and NO oxidation were somewhat recovered., due to the formation of $CaTiO_3$ that weakened the deactivation.

Shen et al. [55] loaded $Ca(NO_3)_2$ with a molar ratio of Ca/Mn = 0.5 on a $Mn-CeO_x/Ti-PILC$ catalyst, and found that the denitration efficiency of the catalyst decreased from 90% before loading to 20% at 180 °C. Zhou et al. [56] used the impregnation method to deposit $CaCl_2$, $CaCO_3$ and $CaSO_4$ on the $Mn-Ce/TiO_2$ catalyst, and the denitration efficiency decreased. At 100 °C, the denitration efficiency of the catalyst without calcium loading reached more than 90%, and the denitration efficiency decreased to 80%, 70% and 40% when $CaCO_3$, $CaSO_4$ and $CaCl_2$ with a mass content of 1% were loaded, respectively. BET, XPS, TPD and other characterizations found that the main reasons for catalyst poisoning were the change of crystal form, the destruction of pore structure, and the reduction of surface active elements and acid sites. Wang et al. [57] studied the effects of $CaCl_2$ and $Ca(OH)_2$ on the denitration efficiency of Mn-Ce-Ti catalysts. The results are shown in Figure 7. At 270 °C, the denitration efficiency of Mn-Ce-Ti catalysts exceeded 90%. After $Ca(OH)_2$ was loaded on the Mn-Ce-Ti catalyst, the denitration efficiency decreased to 75% and 80.8%, respectively.

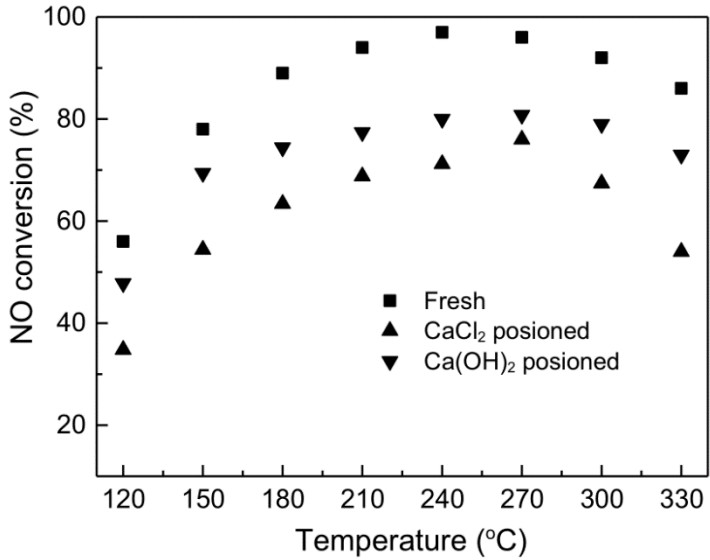

**Figure 7.** NO conversion over fresh and different Ca-poisoned catalysts [57].

In addition, Li et al. [58] also studied the effect of CaO on the $V_2O_5–WO_3/TiO_2$ and $CeO_2-WO_3$ catalyst. The results showed that CW catalyst had a better CaO resistance effect than VWTcatalyst for SCR (Figure 8). At 200 °C, the denitration efficiency of the catalyst after adding 5 wt% CaO was reduced from about 90% without CaO to 50%. XRD Raman(Figure 9), XPS and other analysis showed that CaO inhibited the reducing ability of the catalyst, and significantly reduced the number of Lewis acid and Brønsted acid sites, which reduced the amount of $NH_3$ adsorption of the catalyst. At the same time, as shown in Figure 10, Ca and W formed $CaWO_4$, which reduces the active components on the catalyst surface, thereby reducing the denitration efficiency. Wang et al. [59] found that the calcium poisoning of the $CeO_2-WO_3/TiO_2$ catalyst is due to the fact that $Ca^{2+}$ hinder the conversion between $Ce^{3+}$ and $Ce^{4+}$, reduce the Lewis acid site, inhibit the redox ability and $NH_3$ adsorption, and thus reduce the denitration performance.

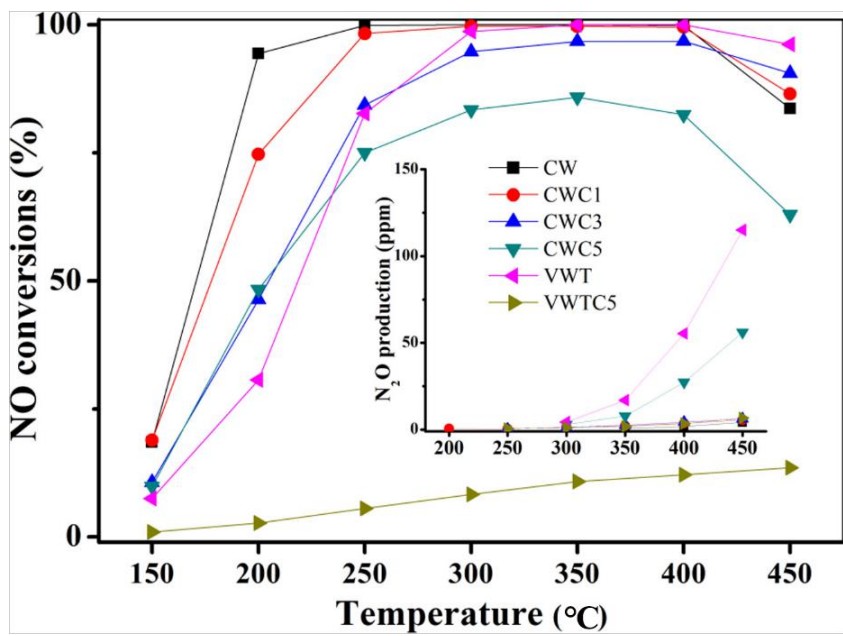

**Figure 8.** NO conversion and N₂O production of CW, VWT and Ca poisoned catalysts. Reaction condition: catalyst amount = 0.1 g, [NO] = [NH₃] = 500 ppm, [O₂] = 3%, N₂ balance, total flow rate = 200 mL/min, GHSV = 120,000 mL/(g·h) [58].

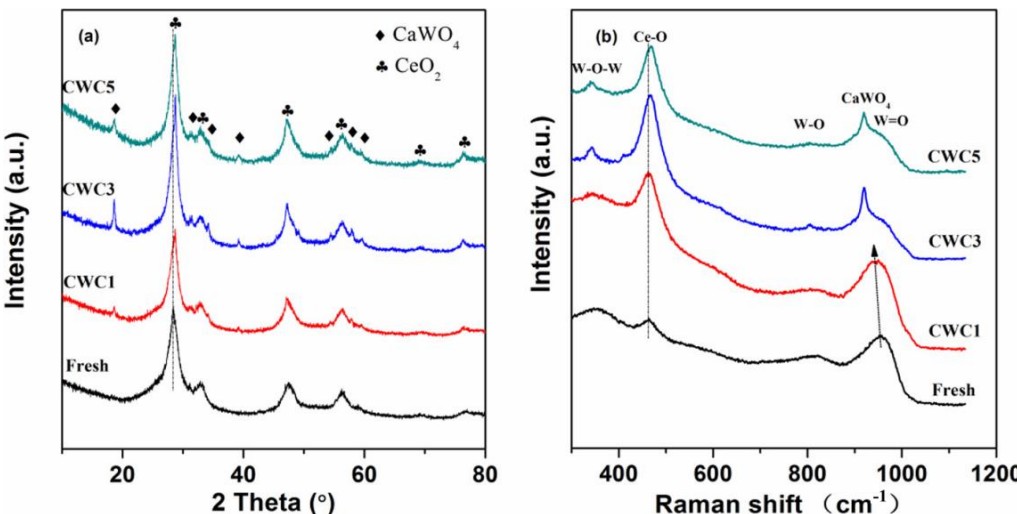

**Figure 9.** XRD patterns (**a**) and Raman spectra (**b**) of fresh and Ca poisoned catalysts [58].

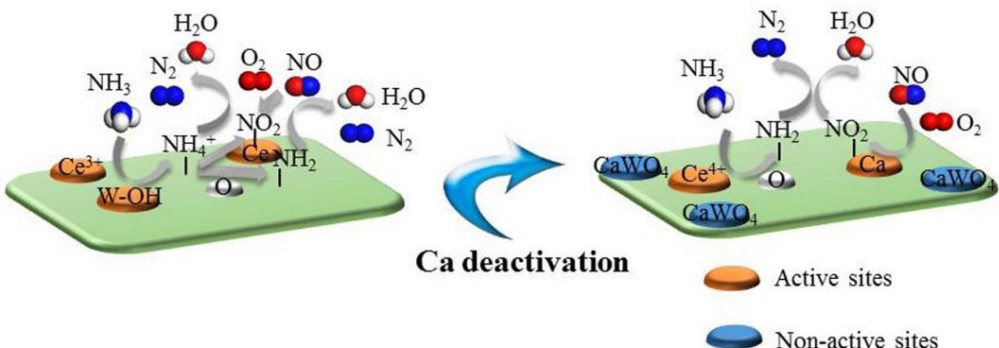

**Figure 10.** Schematic of calcium poisoning mechanism on CeO₂-WO₃ catalyst [58].

Alkaline earth metals in addition to calcium will affect the activity of the catalyst, and Mg will also affect it. Zhou [45] studied the effect of alkaline earth metal Mg deposition

on the denitration performance of Mn-Ce/$TiO_2$ catalysts. The magnesium deposition of different precursors ($MgCO_3$, $MgCl_2$, $Mg(NO_3)_2$, $MgSO_4$) were studied, and it was found that the deposition of different magnesium compounds reduced the catalyst activity compared with the fresh catalyst, and with the increase of the alkaline earth metal Mg loading. The inhibitory effect on catalyst de-stocking activity increased gradually. The deposition of $MgCO_3$ has the strongest inhibitory effect on the catalyst activity, while $Mg(NO_3)_2$ has the weakest effect. It was found by XRD characterization that the deposition of alkaline earth metal Mg would generate $Mn_3O_4$ peaks and $Mn_2O_3$ peaks, which transformed amorphous Mn into crystalline Mn, which was not conducive to the catalytic reaction. It was obtained by BET and $NH_3$-TPD analysis that the deposition of alkaline earth metal Mg would reduce the specific surface area of the catalyst and destroy the Lewis acid sites on the surface of the catalyst. These changes in physical and chemical properties were the main reasons for the decrease of the denitration activity of the catalyst.

### 2.3. The Influence of Pb

Lead is one of the typical heavy metals in the flue gas of coal-fired power plants and municipal solid waste incineration power plants. Lead in fluid gases mainly exists in two forms: particles and gases. Part of the lead is adsorbed or condensed on the surface of the fine particles, and the other part of the lead is converted into lead monoxide or lead chloride in the combustion reaction and enters the atmosphere [60–62]. Studies have shown that the presence of lead has a strong toxicity to SCR catalysts. Measured by Chen et al. [63], the content of Pb in the gas is about 0.072–0.258 µg/$m^3$; the concentration of small particles and gaseous Pb emitted into the atmosphere is relatively high, accounting for 67–81% of the total Pb, and are not captured by dust collectors, which may have a serious impact on SCR catalysts.

Jiang [64] found that when the loading of Pb on the power catalyst reaches 0.19%, the NO conversion rate is only 12%. The low-temperature SCR activity was significantly reduced after doping Pb in Mn-Ce/$TiO_2$ [65]. As shown in Figure 11, when the lead loading is 11%, the NO conversion at 180 °C drops from 100% of the fresh catalyst to 44%. The study of Chen et al. [51] showed that the poisoning effect of lead on SCR catalyst is between the alkali metals potassium and sodium. Guo et al. [66] studied the toxic effects of heavy metals Zn and Pb on the SCR performance of Ce/$TiO_2$ catalysts, and found that the toxic effects of Pb were more serious. The doping of heavy metal Pb will greatly reduce the specific surface area, pore volume, chemical adsorption oxygen content, and surface acidity of the catalyst, as well as increase the crystallinity and grain size of anatase $TiO_2$, resulting in deactivation of the catalyst. In addition, the researchers compared the toxicity of PbO and $PbCl_2$ to certain catalysts. For the $V_2O_5$/$TiO_2$ catalyst, the effect of $PbCl_2$ loading is greater than that of PbO [67,68], the BET specific surface area of the catalyst after $PbCl_2$ loading is smaller, and the acidity and reducibility are also lower. Yet for $CeO_2$-$TiO_2$ catalyst, the result is the opposite, and the effect of PbO generation is greater [69]. From this, it can be concluded that different lead compounds have different effects on different catalysts.

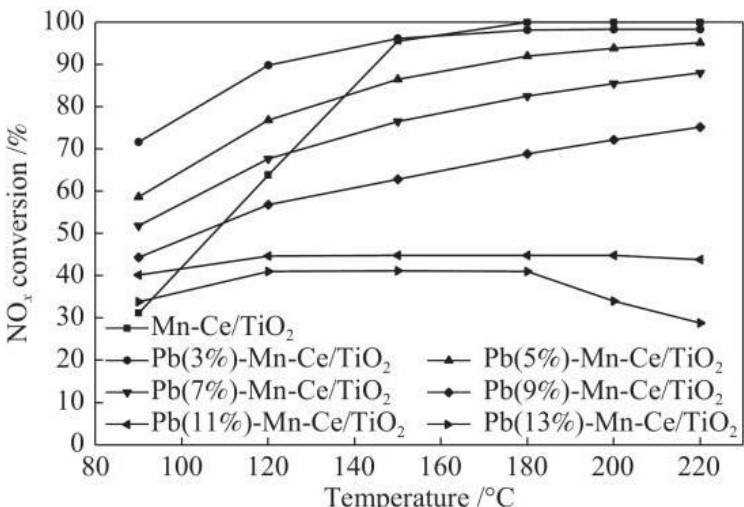

**Figure 11.** Activity of the Pb(x)-Mn-Ce/TiO$_2$ catalysts with different Pb loadings(x) in NH$_3$-SCR of NO reaction conditions: $\varphi$NO$_x$ = 6 × 10$^{-4}$, $\varphi$NH$_3$ = 6.6 × 10$^{-4}$, $\varphi$O$_2$ = 3%~5%, GHSV = 8000 h$^{-1}$ [65].

In order to test the influence of PbO on the performance of catalysts, Zhou et al. [70] synthesized a series of Mn-Ce/TiO$_2$ catalysts doped with PbO by impregnation method. At 200 °C, when the lead-manganese molar ratio reaches 0.5, the NO conversion efficiency of the Mn-Ce/TiO$_2$ catalyst drops from 96.75% to about 40%. The analysis shows that PbO reduces the reducibility, specific surface area, surface Mn$^{4+}$, and Ce$^{3+}$ content, as well as chemisorbed oxygen content of manganese and cerium oxides, resulting in the decrease of SCR performance. The SEM test of Figure 12 showed that the catalyst has obvious aggregation of metal oxides after PbO poisoning. The effect mechanism of PbO on CeO$_2$-MoO$_3$/TiO$_2$ catalyst (as shown in Figure 13) and the reduction in denitration efficiency are due to the formation of a new phase PbMoO$_4$ between PbO and Mo, and this formation inhibits the conversion of surface Ce$^{4+}$ to Ce$^{3+}$, thus significantly reducing the surface acidity and reduction in the catalyst [71]. At the same time, due to the influence of Pb, the amount of chemisorbed oxygen on the surface of the catalyst is reduced, resulting in the inhibition of the NO+O$_2$→NO$_2$ reaction, thereby reducing the activity of the catalyst.

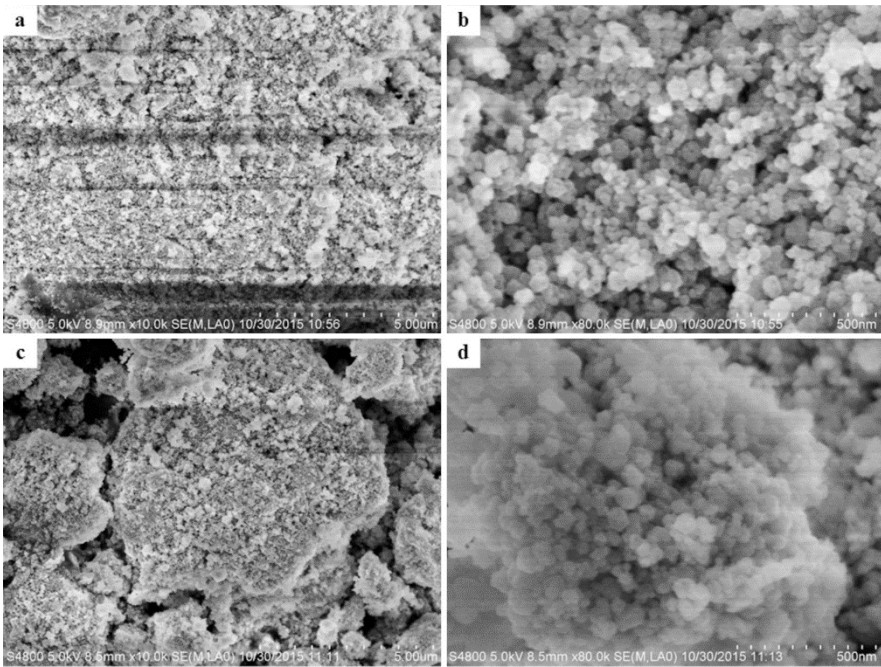

**Figure 12.** SEM images of the catalysts: (**a**) Mn-Ce/TiO₂ 10,000 multiplier; (**b**) Mn-Ce/TiO₂ 80,000 multiplier; (**c**) Pb (0.5)-Mn-Ce/TiO₂ 10,000 multiplier; (**d**) Pb (0.5)-Mn-Ce/TiO₂ 80,000 multiplier [70].

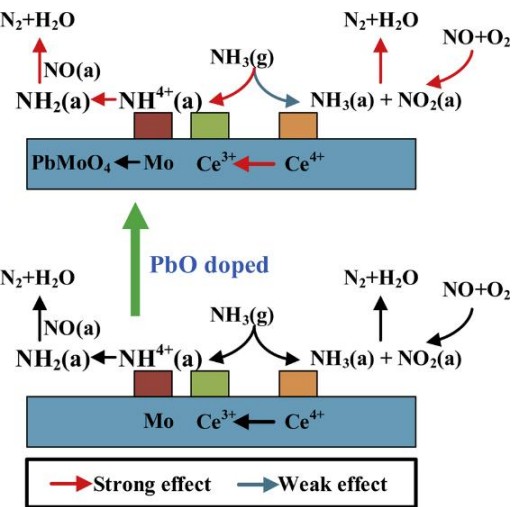

**Figure 13.** Diagram of PbO poisoning on CMT [71].

Studies have shown that PbCl₂ reduces the redox properties and surface acidity of the catalyst, resulting in a decrease in the denitration efficiency. Kong et al. [72] studied the poisoning mechanism of PbO and PbCl₂ on MC catalysts as shown in Figure 14. The toxicity of PbCl₂ is higher than that of PbO, the reason is that PbCl₂ is easier to form crystalline phase, resulting in smaller BET surface area of MC catalyst. At the same time, the newly formed hydrochloric acid preferentially adsorbs on cerium oxide species, forming inactive Cl⁻ bonds and ammonium chloride deposition, which further hinders the conversion of $Ce^{4+}$ to $Ce^{3+}$, and reduces the surface acid sites, resulting in deactivation of the catalyst.

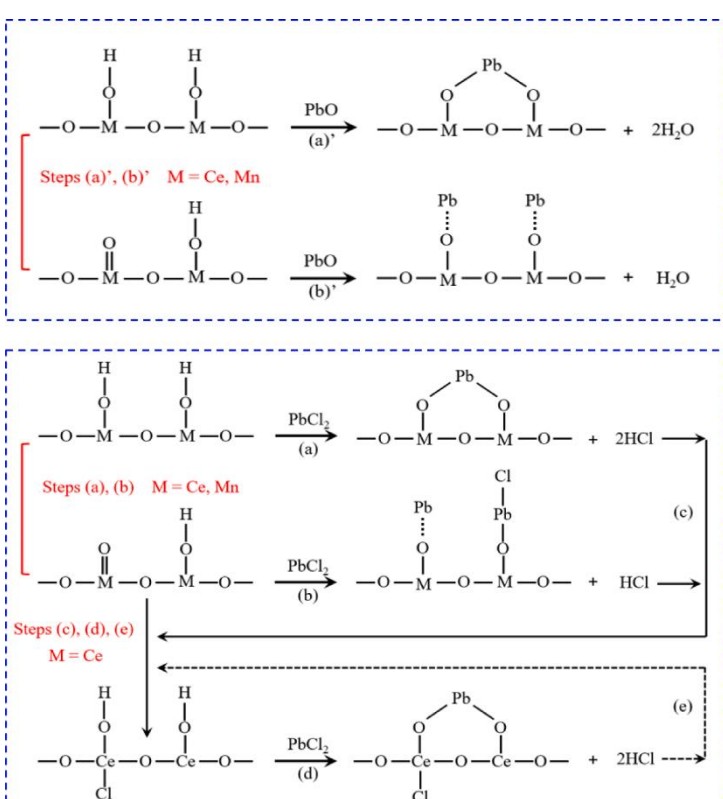

**Figure 14.** Schematic diagram of PbCl₂ and PbO poisoning mechanisms over MC catalysts [72].

Jiang et al. [69] studied the effects of PbO and PbCl$_2$ on the poisoning of the CeO$_2$-TiO$_2$ catalyst in the denitration process. It was found that the NO conversion decreased significantly with the increase of PbO doping amount. When the molar ratio of Pb to Ce exceeds 0.5, the catalyst is almost completely deactivated (Figure 15a). For PbCl$_2$-doped catalysts, PbCl$_2$ has little effect on the catalytic activity when the temperature is lower than 350 °C. In the temperature range of 350~500 °C, with the increase of PbCl$_2$ content, the catalyst showed obvious deactivation (Figure 15b). With the increasing loadings of Pb species, PbO or PbCl$_2$ would gather and form crystallized structure (Figure 16). Combined the XPS, NH$_3$-TPD and H$_2$-TPR tests shown that the PbO doping affects the denitration reaction due to the obvious reduction of the specific BET surface area of the catalyst, thereby reducing the surface Ce$^{3+}$ and chemisorbed oxygen content. PbCl$_2$ reduces the redox properties and surface acidity of the catalyst, and reduces the denitration efficiency. According to DRIFT and other tests, the principle of lead poisoning on CT catalysts (as shown in Figure 17), Pb reduces the Ce$^{3+}$ content on the catalyst surface, which leads to the reduction of Brønsted acid sites, and the reduction of surface chemisorbed oxygen inhibits the progress of NO+O$_2$→NO$_2$ reaction, which are two key factors leading to more severe inactivation of lead oxide.

For deactivated catalysts, nitric acid can be used to restore the redox ability of the catalyst and to increase the surface area and create new acid sites. The use of nitric acid to regenerate Pb-poisoned catalysts can result in almost complete recovery of catalytic activity. Even the catalytic activity exceeds that of fresh catalyst at 80–150 °C.

In conclusion, although cerium-based catalysts have stronger resistance to metal impurities than vanadium-based catalysts, it still affects the SCR activity of catalysts. Therefore, how to improve the resistance of cerium-based catalysts to metal impurities has become the focus of future research. The catalyst can achieve better performance by adjusting the ratio of substances, different synthesis methods, or additives.

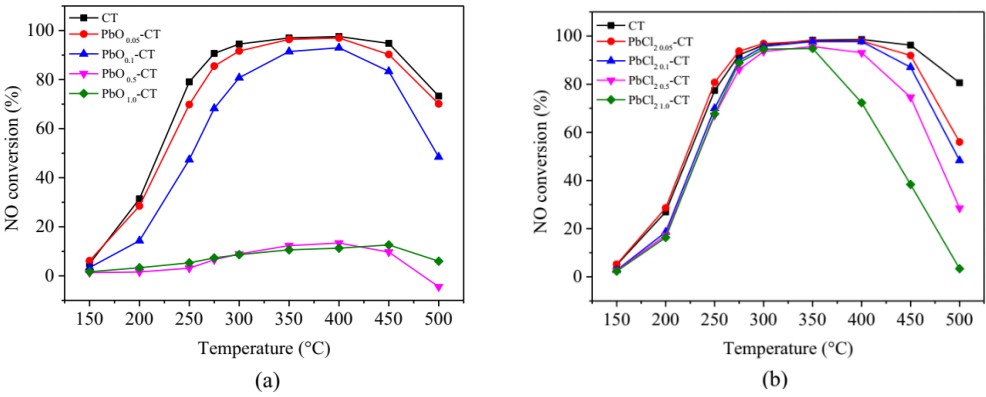

**Figure 15.** NO conversion of PbO$_x$-CT (**a**) and PbCl$_2$ $_x$-CT (**b**). Reaction condition: [NO] = [NH$_3$] = 1000 ppm, [O$_2$] = 3%, N$_2$ balance, total flow rate = 500 mL/min, GHSV = 90,000 h$^{-1}$.[69].

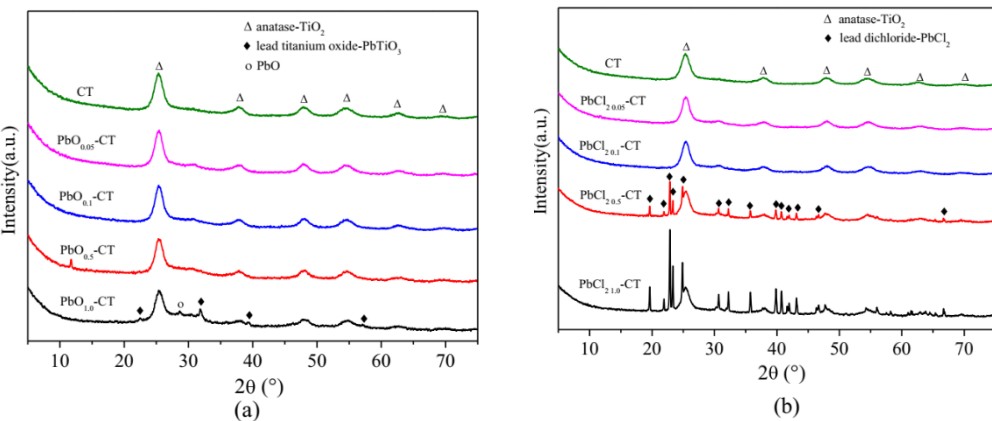

**Figure 16.** XRD patterns of PbOx-CT (**a**) and PbCl2 x-CT (**b**) [69].

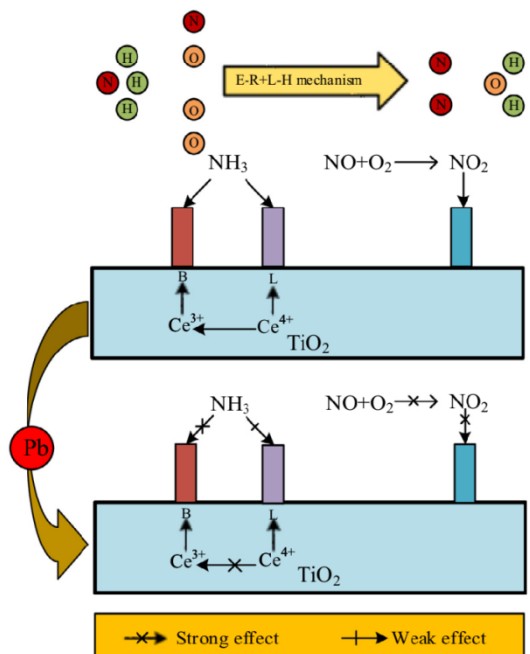

**Figure 17.** Schematic diagram of Pb poisoning on CT [69].

## 3. The Influence of Non-Metallic Impurities

In addition to the influence of metal impurities, the influence of non-metallic impurities in denitration flue gas cannot be ignored, mainly including phosphorus, fluorine, chlorine and sulfur. Their effects on rare earth catalysts are different. The following are mainly analyzed from two aspects: denitration activity and denitration mechanism.

### 3.1. The Effect of Phosphorus

Phosphorus compounds are constituents of dust in fuel gas, and their effect on the catalytic performance of conventional vanadium-based catalysts has been extensively studied. Some literatures point out that P doping can improve the surface acid sites and vanadium species characteristics of $V_2O_5$-$WO_3$/$TiO_2$ catalysts, thereby improving the catalytic activity of the catalysts, but the deposited phosphorus compounds may reduce its catalytic activity, due to the reduced surface active sites and redox properties of the catalyst [53,73–75]. And studies have found that some compounds of phosphorus element have a passivation effect on SCR catalysts, including $H_3PO_4$, $P_2O_5$ and phosphate [76]. Kamata et al. [77] found that the activity of the catalyst decreased with the increase of $P_2O_5$ loading, and the specific surface area and specific pore volume gradually decreased with the increase of surface $P_2O_5$ loading. Kamata et al. also explained that P will replace V and W in V-OH and W-OH to generate P-OH groups. P-OH is not as acidic as V-OH and W-OH, but can provide weaker Brønsted acidic site, so the phosphorus poisoning of the catalyst is not very obvious when the loading is small. In addition, P can also react with the V=O active sites on the catalyst surface to generate substances such as $VOPO_4$, thereby reducing the number of active sites. In light of the above studies, it can be concluded that the content, form and location of phosphorus compounds in the catalyst may have a significant impact on the performance of $NH_3$-SCR catalysts.

However, the effect of phosphorus on the catalytic activity of $NH_3$-SCR ceria-based catalysts remains controversial. To date, only a few studies have reported the effect of phosphorus compounds on the performance of $CeO_2$-$TiO_2$-based catalysts for $NH_3$-SCR. It has been reported that phosphorus doping can significantly improve the pore structure,

thermal stability and surface acidity of TiO₂ [78,79], which in turn affects the performance of CeO₂/TiO₂ catalysts. Yi et al. [80] proposed that depositing phosphorus compounds could enhance the surface acid strength of CeO₂, thereby enhancing its catalytic activity. But a large amount of phosphorus compounds would reduce the redox properties of CeO₂, thereby reducing its catalytic activity. The deposition of phosphorus compounds on the CeO₂-TiO₂ surface could enhance the catalytic activity and resistance to K deactivation of the catalyst, which is due to the enhanced surface acidity and redox properties. The deposited phosphorus compound can deactivate CeO₂-MoO₃/TiO₂, but the formed amorphous CePO₄ species can improve the catalytic performance of CeO₂/TiO₂ catalyst due to the incorporation of phosphorus into CeO₂ [81,82].

Zeng et al. [83] studied the effect of phosphorus on the selective catalytic reduction of NOₓ over CeO₂/TiO₂ catalysts. It was found that phosphorus disrupts the Ti-O-Ce structure due to phosphorus-induced migration of Ti⁴⁺ from the cerium oxide-titanium dioxide solid solution to form a separate titanium dioxide, which promotes the growth of titanium dioxide and cerium oxide grains and reduces the specific surface area of the BET, decreasing the electron transfer capacity and the ratio of Ce³⁺ to surface adsorbed oxygen, resulting in a limitation of the redox performance of the CT catalyst. DRIFT tests showed that phosphorus decreased Lewis acid sites and increased Brønsted acid sites. In addition, phosphorus reduced the adsorption capacity of NOₓ species on the CeO₂/TiO₂ catalyst, changed the adsorption order of NOₓ and ammonia species, and reduced the denitration efficiency.

Cao et al. [84] prepared a phosphorylated CeO₂-TiO₂ catalyst by impregnation method, denoted as xP-CT. The denitration performance test in Figure 18a showed that the catalytic activity of fresh CeO₂-TiO₂ increases rapidly with increasing temperature before 300 °C, and the NOₓ conversion exceeds 85% in the range of 250~400 °C. The activity of the phosphorylation catalyst decreased with the increase of phosphorus loading in the temperature range of 50~300 °C. However, in the high temperature range (above 300 °C), the activity of phosphorus-supported catalysts is higher than that of CeO₂-TiO₂ catalysts. The NOₓ conversion of CeO₂-TiO₂ catalyst at 450 °C is only 14.2%, while the activity of 2.3% P-CT catalyst is still 71.4%. As shown in Figure 19, phosphorus promotes the grain growth of titanium dioxide and ceria in the catalyst, reduces the specific surface area of the catalyst, and inhibits the electron transfer between Cerium and titanium ions, resulting in a decrease in its redox performance. However, when the temperature is above 300 °C, as shown in Figure 18b, P inhibits the NOₓ and N₂O generated by the peroxidation of ammonia gas, thereby improving the activity of the catalyst at high temperature. On the 2.3% P-CT catalyst, the adsorption capacity of ammonia on the Brønsted acid site is greater than that on the Lewis acid site, which also promotes the improvement of the activity at high temperature. The effect of phosphorus on the reaction pathway for NH₃-SCR of NO over the CT catalyst can be depicted as Figure 20. In addition, P does not change the reaction mechanism of the combined action of L-H and E-R on the catalyst surface.

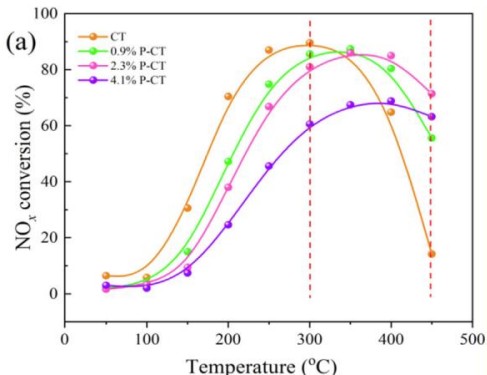
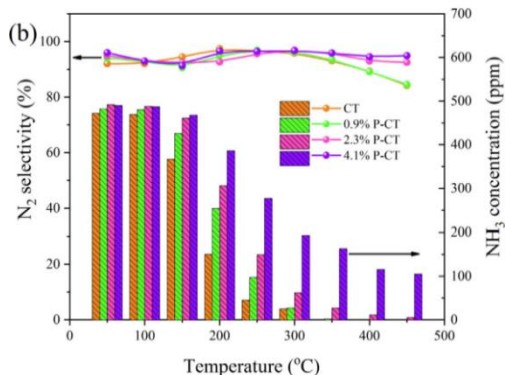

**Figure 18.** (**a**) NOₓ conversion of the CT and phosphorus-loaded catalysts, (**b**) N₂ selectivity and NH₃ concentration during the NH₃-SCR reaction of these catalysts. Reaction conditions: [NH₃] = [NO] = 500 ppm, [O₂] = 5 vol%, N₂ as balance [84].

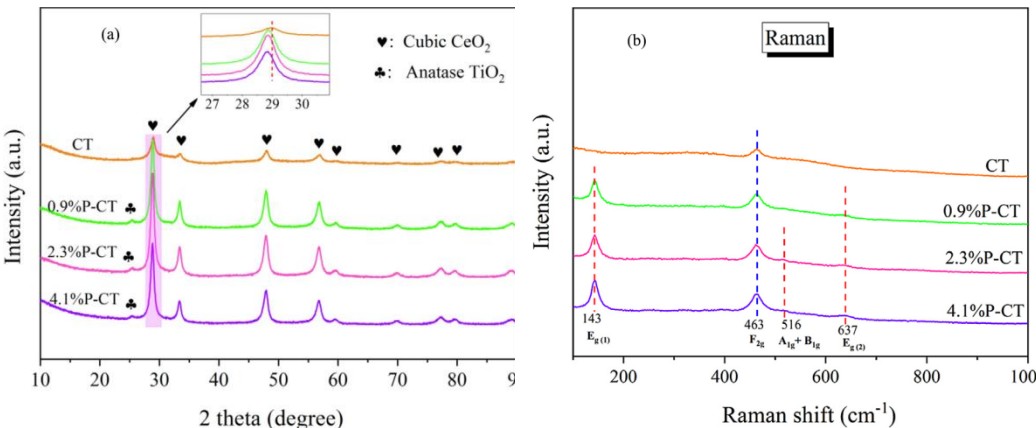

**Figure 19.** (**a**) XRD patterns of CT and phosphorus-loaded catalysts, (**b**) Raman spectra of CT and phosphorus-loaded catalysts [84].

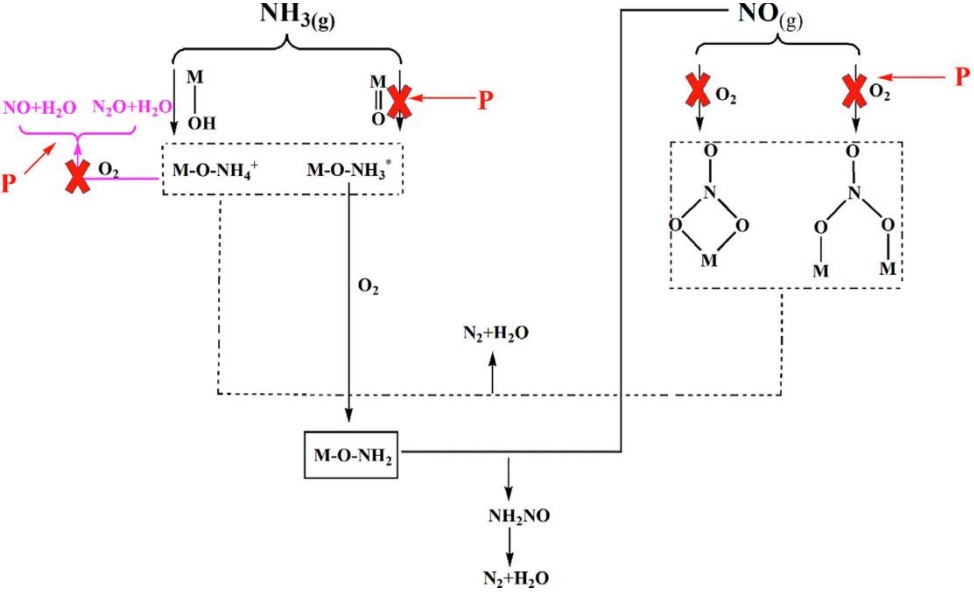

**Figure 20.** The effect of phosphorus on the reaction pathway for NH₃-SCR of NO over CT catalyst [84].

### 3.2. The Effect of Chlorine

As the waste incineration power generation technology is becoming more and more mature, the NOₓ emitted by it needs SCR denitration technology to control. However, in addition to heavy metals and alkali metals, the waste incineration flue gas also contains a certain amount of hydrogen halide gas, and the content of Cl is similar to that of NOₓ, which will have an adverse effect on the activity of the denitration catalyst. At present, the research on heavy metals and alkali metals has received extensive attention, but the research on the effect of Cl on catalyst activity is still lacking.

Over the past decade, some researchers have studied the effects of HCl on SCR catalysts. Lisi et al. [85] studied the effect of HCl on the denitration activity of vanadium-titanium catalyst. The results show that the introduction of HCl will lead to a significant decrease in the activity of the catalyst. The reason is that HCl reacts with the active component vanadium on the surface of the catalyst to generate volatile $VCl_5$, which leads to the decrease of the active component on the surface of the catalyst. In addition, although

HCl gas can form a new acid site on the surface of the catalyst, the performance of the new acid site is lower than that of the original acid site, which eventually leads to the decrease of the activity of the catalyst. However, Hou et al. [86] studied the effect of HCl gas on the denitration activity of $V_2O_5$/AC catalyst. It was found that HCl gas could improve the denitration activity of $V_2O_5$/AC catalyst when the concentration of HCl gas was less than 1200 ppm, the reaction temperature was 120–150 °C and GHSV was less than 6000 $h^{-1}$. This is because $NH_4Cl$ is formed on the surface of the catalyst during the SCR reaction after adding HCl. On the one hand, $NH_4Cl$ can increase the adsorption capacity of $NH_3$ on the surface of the catalyst, and on the other hand, $NH_4Cl$ can also react with NO to avoid the continuous accumulation of $NH_4Cl$ on the surface of the catalyst, so that the SCR reaction continues well. However, when the concentration of HCl gas was increased or the reaction temperature was changed, HCl gas resulted in a significant decrease in the denitration activity of the catalyst.

It is known that the addition of cerium in the catalyst can effectively resist the influence of Cl; the study by Jin et al. [87] had shown that HCl will react with Cu ions in the CuHM catalyst to form $Cu_2Cl(OH)_3$, resulting in the change of valence state or phase of Cu ions on the catalyst, reducing the content of Cu on the surface of the catalyst, resulting in the decrease of catalyst activity. However, when Ce was added to the CuHM zeolite catalyst, the resistance of the obtained CeCuHM catalyst to HCl gas was improved, because the addition of Ce could not only reduce the loss of Cu ions on the catalyst, but also inhibit the transformation of $Cu^{2+}$ to $Cu^+$ [88].

Yang et al. [89] studied the effect of $Cl^-$ on the denitration activity of $Ce/TiO_2$ catalysts in the temperature range of 75~225 °C, and the results showed that the addition of $Cl^-$ would inhibit the adsorption of $NH_3$ and $NO_x$ on the surface of the catalyst, which is not conducive to the whole SCR reaction, and ultimately led to the decrease of the denitration activity of $Ce/TiO_2$, as shown in Figures 21 and 22. Chang et al. [90] studied the effect of $SO_2$ and HCl gas coexistence on $Rh/Al_2O_3$ catalyst and found that there is competition between $SO_2$ gas and HCl gas adsorption on the catalyst surface. When 500 ppm $SO_2$ and 500 ppm HCl gas were introduced into the SCR reaction process, the catalyst was completely deactivated.

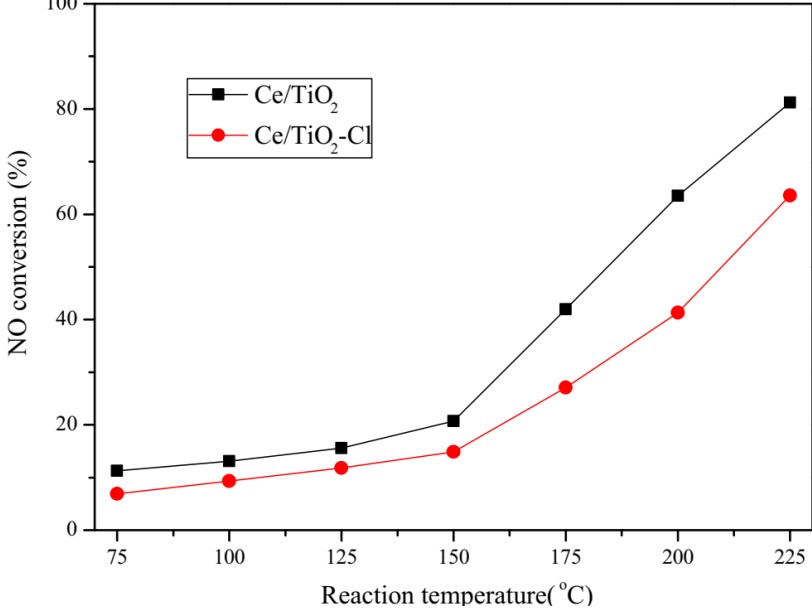

**Figure 21.** The SCR activities of $Ce/TiO_2$ and $Ce/TiO_2$-Cl as a function of reaction temperature Reaction conditions: [NO] = [$NH_3$] = 600 ppm, [$O_2$] = 5%, balance Ar, GHSV = 108,000 $h^{-1}$ [89].

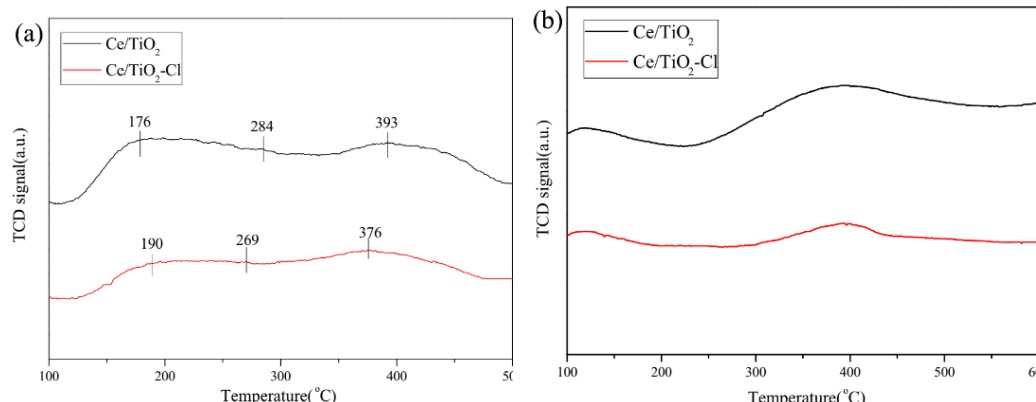

**Figure 22.** (**a**) NH₃-TPD profiles of the two catalyst samples, (**b**) NO-TPD profiles of the two catalyst samples [89].

Lu's research [91] found that HCl gas has an inhibitory effect on the SCR denitration activity of $CeO_2/TiO_2$ and $CeO_2$-$MoO_3/TiO_2$ catalysts. In the temperature test range of 150~500 °C, the denitration efficiency of the catalysts are all significantly decreased, and the temperature window was greatly reduced. Although HCl inhibits the catalyst obviously, the $CeO_2$-$MoO_3/TiO_2$ catalyst after HCl gas treatment still maintains a NO conversion rate higher than 90% in the range of 400~450 °C. HCl led to the decrease of specific surface area, the increase of crystallinity, the decrease of redox capacity and the substantial decrease of surface acid sites of the catalyst, which further affected the activity of the catalyst. In the temperature range from 150 to 300 °C, the effect of HCl on the activity of CMT is more serious than that of CT, because the reduction of acid sites corresponding to CMT is more severe.

In light of the above research results, it can be found that Cl has different effects on the denitration activities of different types of SCR catalysts, and there are also great differences in the activity changes of the same catalyst under different reaction temperatures and different concentrations of HCl gas.

*3.3. Influence of Fluorine*

In the metallurgical industry, the raw minerals and coal used for production often contain fluorine, which is emitted with the flue gas during the smelting process. The fluorine in the flue gas has great damage to the anti-corrosion layer of the flue, and many studies have proved that the fluoride additive can enhance the catalytic activity of NH₃-SCR. Studies have shown that F-doped $V_2O_5$–$WO_3/TiO_2$ catalysts exhibit high activity for NH₃ low-temperature SCR. F doping improves the oxygen vacancy interaction between $WO_3$ and $TiO_2$, resulting in the increase of superoxide ions in chemisorbed oxygen and NO oxidation, which is of great significance for low-temperature SCR reactions [92]. For $V_2O_5/TiO_2$ catalysts, F doping improves the interaction of V species with $TiO_2$ via oxygen vacancies and electrons, which significantly promotes low-temperature SCR activity [93]. For traditional vanadium-based catalysts, there are many studies on F-doping, but the research on its effect on rare earth denitration catalysts is not yet in-depth.

Zhang et al. [94] found that F doping would promote the low-temperature SCR activity of $CeO_2$-$TiO_2$ catalysts. At 180 °C, the NO conversion rate of $Ce_{0.3}TiF_{1.5}$ reached 92.19%, showing excellent catalytic performance. Through BET, XRD, PL spectroscopy, Raman spectroscopy and XPS analysis, it is known that F-doped $CeO_2$-$TiO_2$ catalyst can inhibit crystallization, make the catalyst have a better amorphous structure, and increase the active sites on the surface of the catalyst. A stronger interaction occurs between Ce and Ti, which is favorable for electron transfer, increases oxygen vacancies and chemisorbed oxygen, and improves the morphology of $Ce^{3+}$, thereby promoting the catalytic

performance. NH$_3$-TPD analysis showed that a moderate amount of F doping can significantly increase the number of acid sites on the catalyst surface, especially Lewis acid sites, which are related to the higher chemisorbed oxygen on the catalyst surface. The DRIFTS results show that the doping of F can promote the reaction of superoxide radicals (O$_2^-$) on the catalyst surface with NO in the gas phase to generate nitro (NO$_3^-$) and nitroso (NO$_2^-$) species. These species are the intermediate products of the reaction with the reducing gas NH$_3$ in the reaction gas. The increase of intermediate species can speed up the SCR reaction process, thereby improving the denitration activity [11,95,96].

In order to explore the effect of the preparation method of the catalyst on the F-doped CeO$_2$/TiO$_2$ catalyst, Zhang [94,97] also tested the performance of the F-doped catalyst by the sol-gel-impregnation method and the co-precipitation method, respectively. The results show that the catalyst prepared by the sol-gel-impregnation method can improve the denitration performance after F doping, but the conversion rate of NO$_x$ is still lower than 55% in the low temperature region below 210 °C, which has no practical significance. However, the catalyst prepared by the coprecipitation method showed excellent performance, and its sulfur resistance and water resistance were also enhanced. When the space velocity is 28,000 h$^{-1}$ and the reaction temperature is 210 °C, the denitration efficiency of the Ce$_{0.3}$TiF$_{1.5}$ catalyst is almost 100%, and when the space velocity is 41,000 h$^{-1}$, the catalyst can still obtain 95% denitration at the reaction temperature of 210 °C efficiency, as shown in Figure 23. The F-doped cerium-titanium catalysts prepared by the co-precipitation method all showed an amorphous structure with high redox ability, Figure 24 indicated that F-doping resulted in more oxygen vacancies, especially the number of single-electron-trapped oxygen vacancy (F$^+$ center). Oxygen vacancies could absorb O$_2$ to form chemisorbed oxygen. F-doping might enhance the interaction between titanium and cerium, which was in good agreement with the XRD results in Figure 25. These are all important factors for the improvement of catalyst activity.

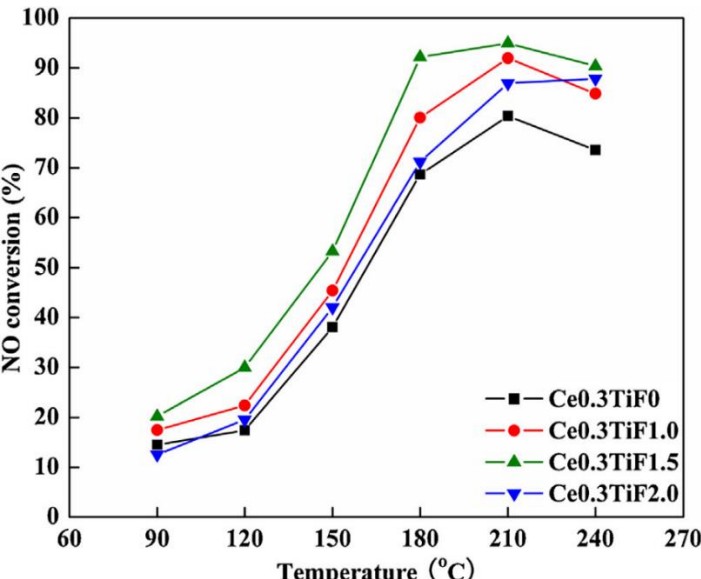

**Figure 23.** NO$_x$ conversion of Ce$_{0.3}$TiF$_y$ catalyst with different F doping Reaction conditions: 100 mL/min, 0.05% NO, 0.06% NH$_3$, 5 vol% O$_2$, N$_2$ balance, GHSV = 28,000 h$^{-1}$ [94].

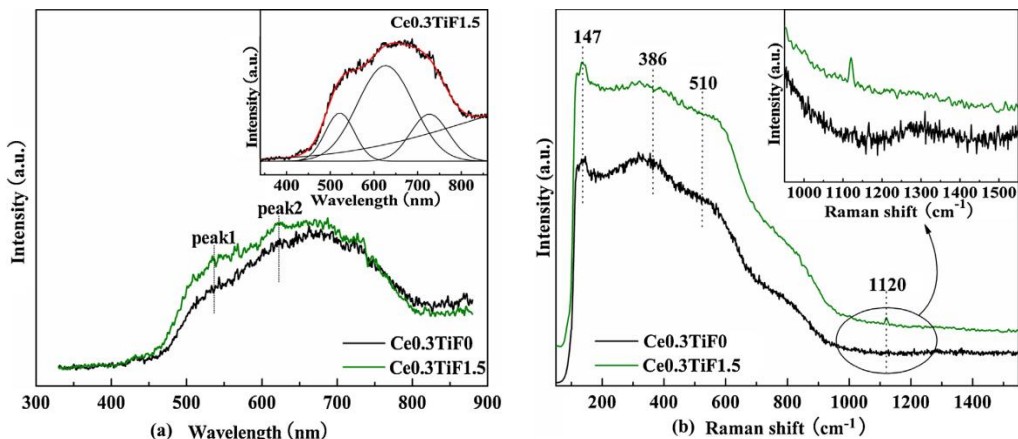

**Figure 24.** (**a**) PL spectra of $Ce_{0.3}TiF_0$ and $Ce_{0.3}TiF_{1.5}$ samples, (**b**) Raman spectra of $Ce_{0.3}TiF_0$ and $Ce_{0.3}TiF_{1.5}$ samples [94].

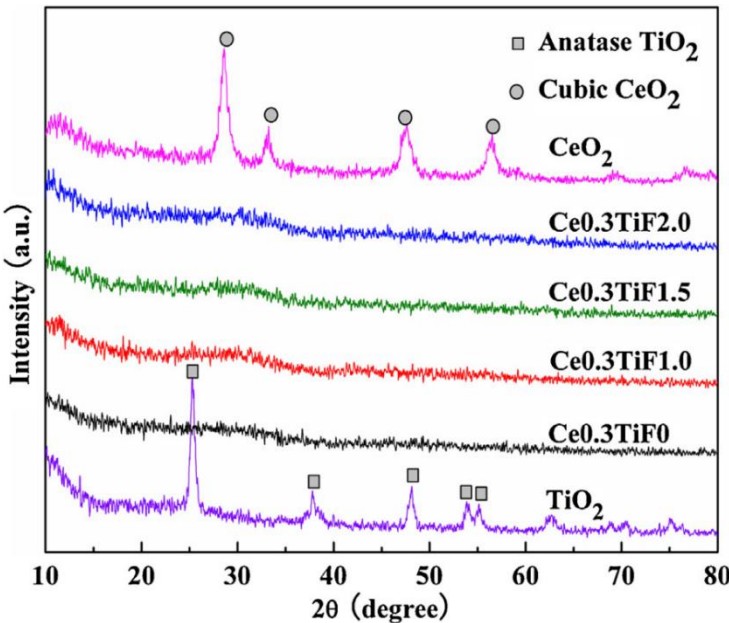

**Figure 25.** X-ray diffraction patterns of the $Ce_{0.3}TiF_y$, $TiO_2$ and $CeO_2$ [94].

In addition, some scholars have also studied the effect of HF on the denitration performance of SCR catalysts. For example, Yang et al. [98] studied the effect of HF treatment on the SCR performance of $CeO_2$ catalysts. The experimental results show that HF treatment can greatly enhance the SCR activity of $CeO_2$ catalysts. From the characterization results, it can be found that HF treatment of $CeO_2$ catalyst will lead to lower crystallinity, better reducibility, stronger $NH_3$ adsorption capacity, and more surface adsorption of oxygen, all of which will lead to enhanced catalyst activity. The $CeO_2$ was treated with HF gas, and the treated $CeO_2$ showed good denitration activity in the range of 100–400 °C. In addition, Jin et al. [99] obtained similar results when they studied $CeO_2(ZrO_2)/TiO_2$ catalysts modified by HF solution. The addition of HF improved the oxygen storage capacity of the catalysts. Figure 26 showed HR-TEM images of $TiO_2$-0F and $TiO_2$-10F. The average size of $TiO_2$-10F (15–20 nm) was much larger than that of $TiO_2$-0F (5–10 nm) In addition, the lattice fringes with an interplanar spacing of 0.35 nm and 0.235 nm were consistent with the d-spacing of (1 0 1) and (0 0 1) facets, respectively. The synergistic effect of (1 0 1) and (0 0 1) crystal planes and the increase of surface chemisorbed oxygen and $Ce^{3+}$ concentrations are beneficial to the improvement of catalytic activity, as shown in Figure 27.

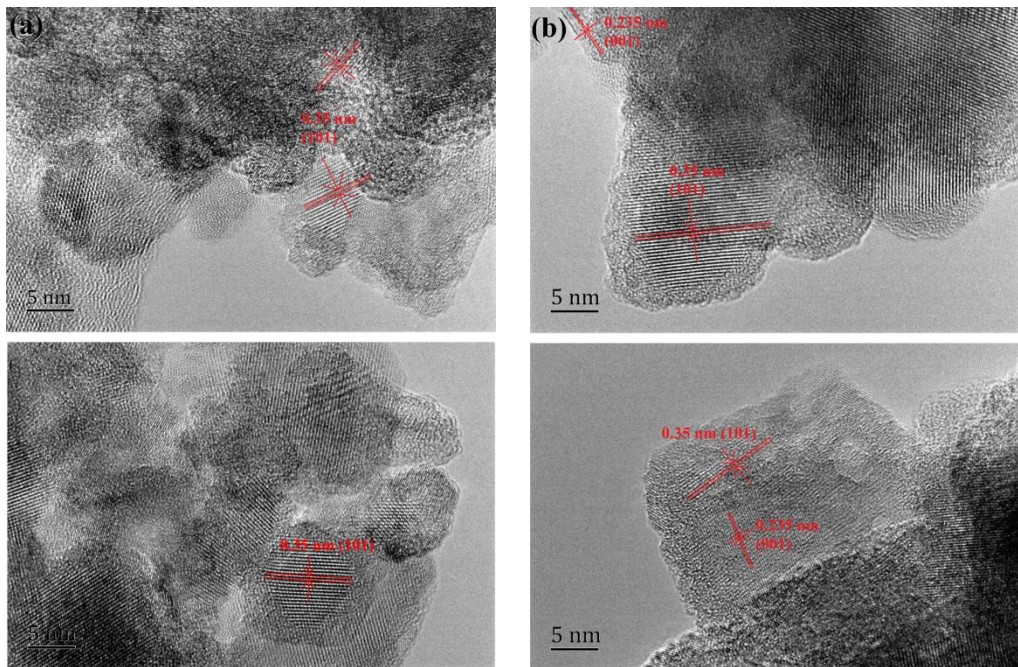

**Figure 26.** HR-TEM images of TiO$_2$-0F (**a**), TiO$_2$-10F (**b**) [99].

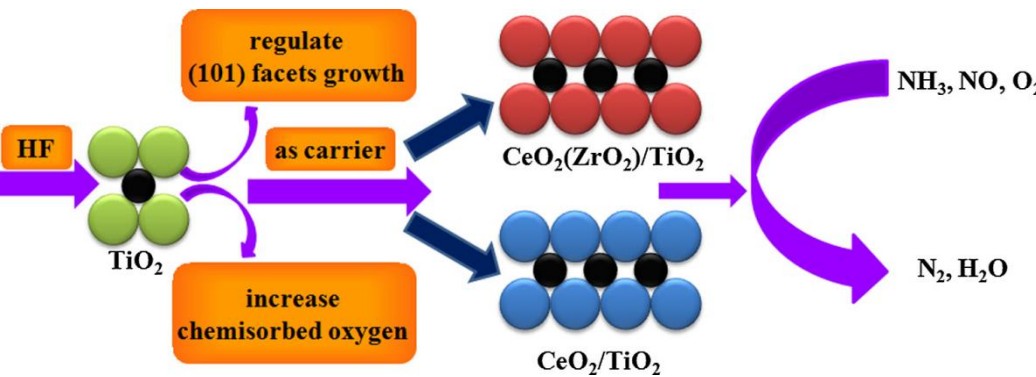

**Figure 27.** Influence mechanism of HF on CeO$_2$(ZrO$_2$)/TiO$_2$ catalysts [99].

In conclusion, appropriate amounts of fluorine and hydrogen fluoride can improve the denitration activity of rare earth catalysts, and can also appropriately improve sulfur resistance and water resistance, and are also closely related to the preparation method of the catalyst.

### *3.4. The Effect of Sulfur*

It has been pointed out that transition metal oxides (eg, VO$_x$ [100], MnO$_x$ [101], CeO$_x$ [102], CuO$_x$ [103] and FeO$_x$ [104], etc.) are the main active components of low-temperature SCR denitration catalysts, although low-temperature SCR catalysts exhibit excellent low-temperature SCR activity [105–107], most fossil fuels contain sulfur, resulting in large amounts of SO$_2$ in exhaust gas, these traditional catalysts have poor resistance to SO$_2$, and even lead to deactivation directly. The resistance of denitration catalysts to SO$_2$ is an important characterization of catalyst performance. In general, the influence of SO$_2$ on the catalyst is mainly manifested in two aspects: first, SO$_2$ in the flue gas will react with ammonia to form sulfates, such as (NH$_4$)$_2$SO$_3$ and NH$_4$HSO$_4$, which do not decompose at low temperatures and eventually deposit on the catalyst surface, the specific surface area of the catalyst is reduced and the active site of the SCR catalyst is blocked. Second, SO$_2$ will compete with NH$_3$ for adsorption and sulfate, as well as the surface-active substances, thereby inhibiting the activity of the catalyst [108,109]. Therefore, research on improving

the sulfur tolerance of low-temperature SCR catalysts has received extensive attention. Some studies have found that doping rare earth elements (Ce [110], Pr [111], Sm [112], and Eu [113], etc.) can effectively improve the resistance of catalysts to $SO_2$. $CeO_2$ is often selected as the promoter or active component of SCR catalyst because of its excellent oxygen storage/release ability and strong redox performance. The addition of cerium has been reported to alleviate the sulfation of catalyst active sites and the formation of ammonium sulfate [108], promoting the anti-sulfur properties of SCR catalysts. However, the effect of $SO_2$ on the activity of cerium-based catalysts and the specific mechanism still need to be further explored.

Sheng et al. [114] showed that $SO_2$ can form $Mn(SO_4)_2$ and $Ce_2(SO_4)_3$ with $MnO_x$ and $CeO_2$ in Mn-Ce/$TiO_2$ catalysts, resulting in the decrease of catalyst activity. Jin et al. [18] also studied the effect of Mn-Ce/$TiO_2$ catalyst on $SO_2$ tolerance, as shown in Figure 28, and found that under the same reaction conditions in $SO_2$ atmosphere, Mn/$TiO_2$ catalyst only retained 25% of NO conversion, while the Mn-Ce/$TiO_2$ catalyst retained about 60% NO conversion. In-situ DRIFT analysis (Figure 29) found that the formed sulfate species on Mn-Ce/$TiO_2$ surface decomposed much more easily than those on Mn/$TiO_2$ surface. The lower thermal stability of the sulfation species on Mn-Ce/$TiO_2$ may lead to an increase in its sulfur tolerance. In the presence of $SO_2$, sulfate species can be preferentially formed on the Ce dopant, less sulfonation of the main active phase $MnO_x$, and retention of some Lewis acid sites on $MnO_x$ (mechanism Figure 30) to meet the low temperature SCR cycle. The calculation of the exchange correlation function between VASP4.6 and GGA+PW91 [115] showed that the doping of Ce reduced the binding energy of ammonium and sulfate ions, thus making ammonium sulfate easier to decompose. TG-DSC results also confirmed that the decomposition temperature of $NH_4HSO_4$ on Mn-Ce/$TiO_2$ is about 70 °C lower than that on Mn/$TiO_2$. In addition, Gu et al. [116] also found that the surface sulfonation of $CeO_2$ can improve the SCR activity. Wang et al. [110] also found that the formation of $MnCeO_x$ solid solution and the preferential sulfation of $CeO_2$ make the MnCe/Ti catalyst have higher SCR activity and stronger resistance to $SO_2$ performance. These results indicate that Ce doping can effectively delay the formation of sulfated species on the surface, thereby improving the sulfur tolerance of Ce-modified catalysts, so rare earth catalysts have better resistance to $SO_2$ than traditional catalysts.

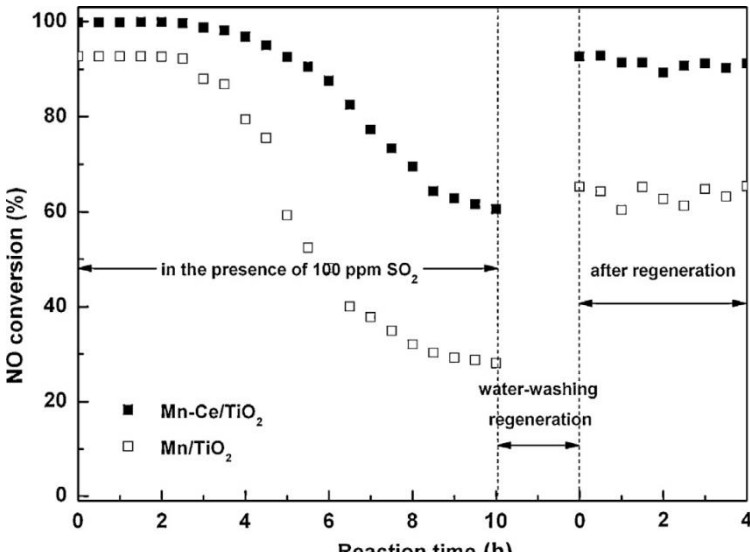

**Figure 28.** SCR activities of Mn/$TiO_2$ and Mn-Ce/$TiO_2$ in the presence of $SO_2$ at 150 °C. ([$NH_3$] = [NO] = 800 ppm, [$O_2$] = 3%, [$SO_2$] = 100 ppm, [$H_2O$] = 3 vol%, $N_2$ balance, GHSV = 40,000 $h^{-1}$) [18].

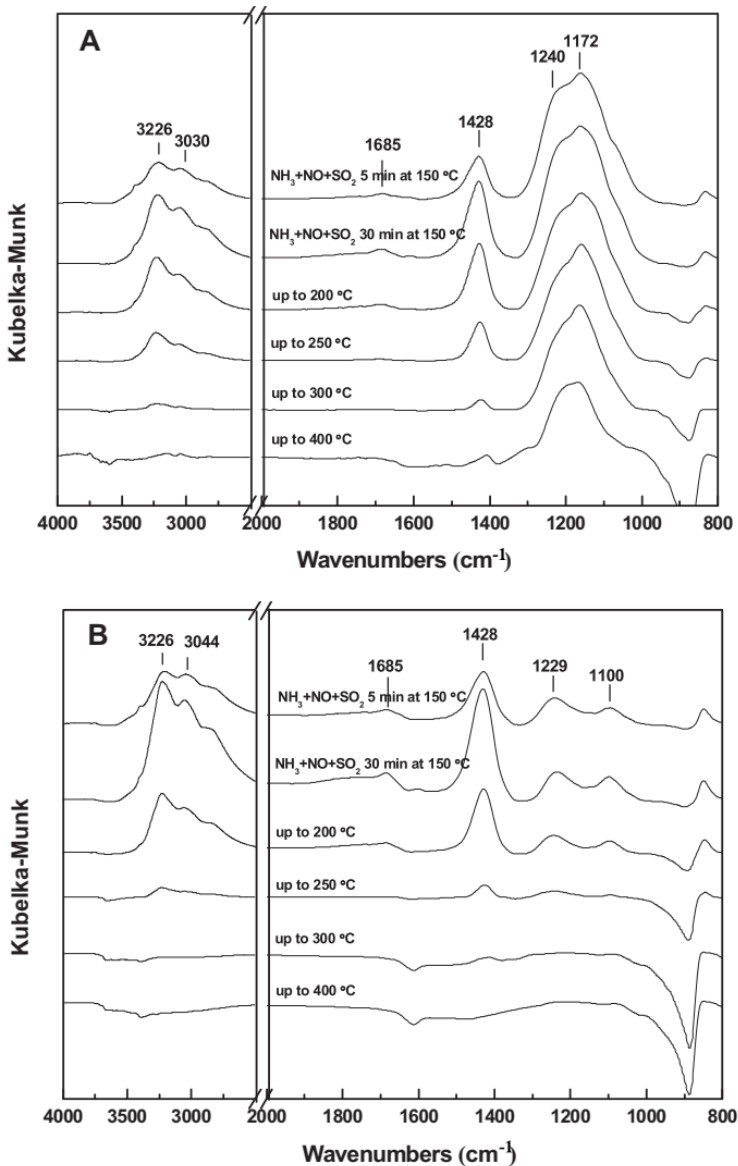

**Figure 29.** DRIFT spectra of Mn/TiO$_2$ (**A**) and Mn-Ce/TiO$_2$ (**B**) exposed to 800 ppm. NH$_3$ + 800 ppm NO + 100 ppm SO$_2$ in the presence of O$_2$ for various times at 150 °C. Subsequently, the atmosphere was switched to only He and the temperature was escalated to 400 °C [18].

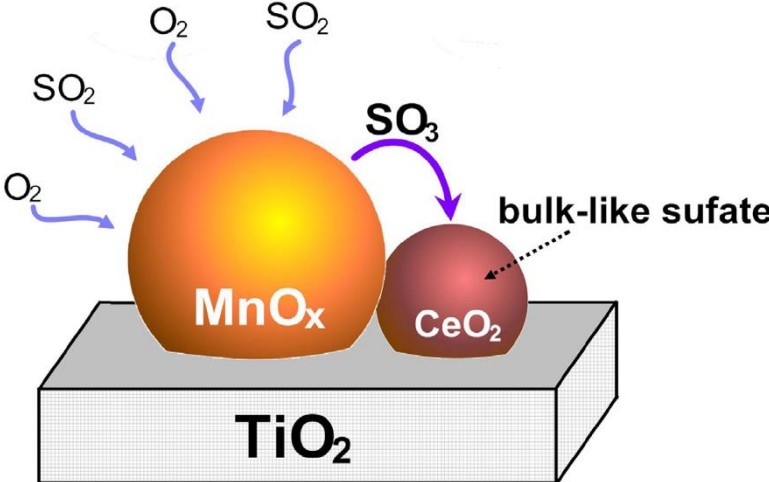

**Figure 30.** The formation pathway of bulk-like sulfate on Mn-Ce/TiO$_2$ samples [18].

It has also been suggested that the addition of modifiers to cerium-based catalysts can further improve the sulfur tolerance of the catalysts. For example, Shan et al. [23] added $WO_3$ to $CeO_2$-$TiO_2$ to form $Ce_{0.2}W_{0.2}TiO_x$. This catalyst maintained a $NO_x$ conversion rate of nearly 100% in the presence of 100 ppm $SO_2$ at a temperature of 300 °C. Shen et al. [117] also found that the zirconium additive had a similar promoting effect on the catalytic performance of $Ti_{0.8}Ce_{0.2}O_2$. Iron doping also has a positive effect on the $SO_2$ tolerance of the Mn-Ce/$TiO_2$ catalyst, as iron oxides significantly reduce the sulfate formation rate [118]. Liu et al. [119] reported that Ce/$TiO_2$-$SiO_2$ had stronger $SO_2$ resistance than Ce/$TiO_2$, and the study showed that the introduction of $SiO_2$ further weakened the basicity of the Ce/$TiO_2$–$SiO_2$ catalyst surface. Compared with Ce/$TiO_2$, Ce/$TiO_2$-$SiO_2$ has less sulfate accumulation on the surface. However, Yu et al. [120] believed that in the SCR reaction of NO and $NH_3$ at low temperature, the catalyst structure rather than the catalyst composition determines the ability of the catalyst to resist $SO_2$ poisoning. Furthermore, the mesoporous structure promotes $SO_2$ resistance compared with the microporous structure. There are also studies showing that the preparation method also affects the resistance of $CeO_2$-$TiO_2$ catalysts to $SO_2$. For example, the samples prepared by the sol-gel method by Gao et al. [20] exhibited better $SO_2$ resistance than the samples prepared by the impregnation method and co-precipitation method. In addition, Shan et al. [121] reported that on Ce-Ti mixed oxides prepared by uniform precipitation, the NO conversion was almost unchanged at 300 °C with the addition of 100 ppm $SO_2$ for 24 h. Therefore, the resistance of cerium-based catalysts to $SO_2$ is not only related to the composition and structure of the catalyst but also to the preparation method.

Although cerium-based catalysts have good resistance to $SO_2$, the research on catalyst deactivation regeneration is still important due to the different application environments. A considerable part of the literature studies the regeneration of deactivated catalysts, and there are many methods to regenerate deactivated catalysts, such as water washing, thermal regeneration, and reductive regeneration. The study by Sheng et al. [114] found that water washing has the best regeneration performance for toxic catalysts, especially under the action of ultrasonic vibration, the catalytic activity can recover to 91.3%, as shown in Figure 31, almost reaching the level of fresh catalysts. Other studies have drawn similar conclusions: after deionized water washing and regeneration, it was found that the NO conversion rate of the Mn-Ce/$TiO_2$ catalyst could be restored to more than 90%, while the NO conversion rate of the Mn/$TiO_2$ catalyst was only restored to 60% [18]. This is because the $SO_2$ deactivation mechanism of the $NH_3$-SCR catalyst is due to active phase sulfation and surface ammonium sulfate/bisulfate deposition, which can be easily removed by water washing. And the main washing products Nitrate $NO_3^-$, Sulfate $SO_4^{2-}$, and ammonium $NH_4^+$ can be recycled to improve the economic benefits of the low temperature SCR technology.

Wang et al. [122] also found that although the coexistence of $H_2O$ and $SO_2$ aggravated the deactivation of the catalyst, the surface hydroxylation of the catalyst prevented the metal sulfation and significantly alleviated the irreversible poisoning. Thermal treatment with $H_2O$ or $O_2$ has been proven can regenerate the $SO_2$ poisoned catalyst effectively, for both operations facilitate the decomposition of the deposited $(NH_4)_2SO_4$ or $NH_4HSO_4$ and induce the sub-bulk/bulk S atom out-migration.

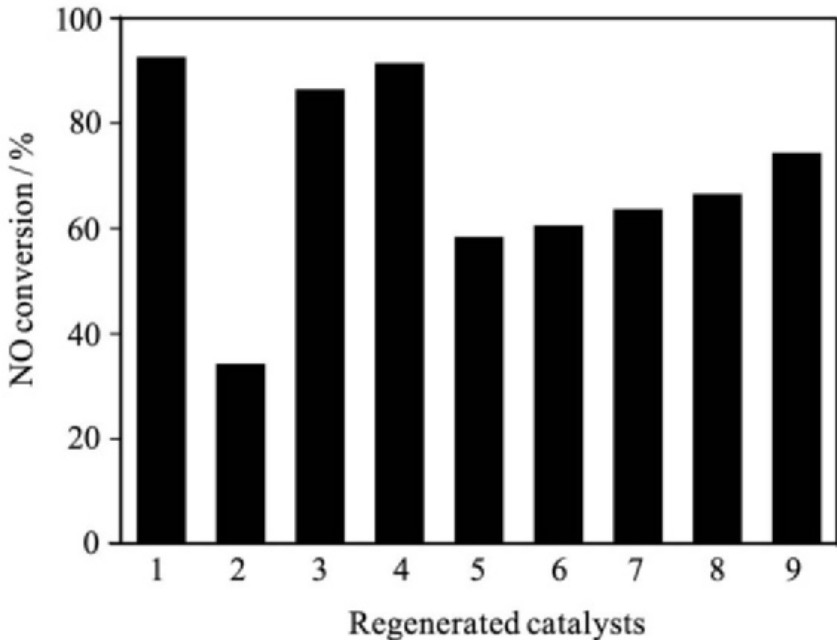

**Figure 31.** SCR activities of fresh Mn-Ce/TiO$_2$ (0.075) (1), deactivated sample (2), and regenerated catalysts treated by water washing (3), water washing with ultrasonic vibration (4), heating in air (5), heating in N$_2$ (6), heating in Ar (7), H$_2$ redution (8), and NH$_3$ reduction (9) (operating conditions: 600 ppm NO, 600 ppm NH$_3$, 3% O$_2$, 3% H$_2$O, and balance N$_2$, GHSV = 40,000 h$^{-1}$, 120 °C reaction temperature) [114].

In short, non-metallic impurities widely exist in denitration flue gas, and the harm to the catalyst is inevitable. Rare earth catalysts with excellent performance and good resistance to impurities are bound to become the focus of industry research in the future, some people have achieved good results in this regard. Although it has excellent performance, some aspects need to be further explored. It is urgent to explore new synthesis methods and new material ratio.

Finally, the effects of different types of impurities on rare earth catalysts are summarized in Table 1.

**Table 1.** Effects of Various Impurities on Rare Earth Catalysts.

| Types of Impurities | Effect on Rare Earth Catalysts |
| --- | --- |
| Na, K | K and Na will decrease the acid sites of the catalyst, and their oxides and chlorides will weaken the reaction activity of the catalyst surface, inhibit the formation of oxygen vacancies and chemical adsorbed oxygen, thereby reducing the NH$_3$ adsorption amount and weakening the denitrification performance. The influence of K is greater than that of Na, and the influence of oxides is more serious than that of chlorides. |
| Ca | Ca deposition can destroy the pore structure of the catalyst, reduce the surface active elements and acid sites, and Mg has a similar effect. |
| Pb | Lead will reduce the redox performance, chemical adsorption of oxygen and specific surface area of the catalyst. The toxicity of lead chloride is higher than lead oxide, because lead chloride is easier to form crystalline phase. |
| P | At low temperature, P promotes the grain growth of TiO$_2$ and CeO$_2$ in the catalyst, reduces the specific surface area of the catalyst, inhibits the electron transfer between Ce and Ti ions, and reduces its redox performance. At high temperature, P inhibits NO$_x$ and N$_2$O produced by ammonia peroxidation, thereby increasing its activity. |

| | |
|---|---|
| Cl | HCl led to the decrease of specific surface area, the increase of crystallinity, the decrease of redox ability, and the significant decrease of surface acid sites, which further affected the catalyst activity. |
| F | F can inhibit crystallization, so that the catalyst has more surface active sites, increasing oxygen vacancies and chemisorption oxygen. In addition, the addition of F can bring more $NO_x$ adsorption sites and the formation of intermediate species, thereby promoting the activity of the catalyst. |
| S | Ce in rare earth catalysts can effectively delay the formation of surface sulfating substances, reduce the binding energy of ammonium and sulfate ions, so that ammonium sulfate is easier to decompose and improve the sulfur resistance of Ce modified catalysts. |

## 4. Conclusions and Perspectives

In conclusion, cerium-based catalysts exhibit good denitration activity and have been widely studied, but the composition of denitration flue gas is complex, and the influence on catalyst activity is unavoidable. Alkali metals will reduce the acid sites of the catalyst and reduce the amount of $NH_3$ gas adsorption; the deposition of alkaline earth metals will destroy the pore structure of the catalyst; lead will reduce the redox performance and specific surface area of the catalyst; P and Cl will promote grain growth and lead to increased crystallinity. However, F can inhibit crystallization, increase oxygen vacancies and chemical adsorption of oxygen, make the catalyst have more surface-active sites, and promote the formation of intermediate substances, improving the activity of the catalyst. Compared with vanadium-based catalyst, the great oxygen storage performance and excellent redox performance of $CeO_2$ make cerium-based catalysts more resistant to various impurities, especially to $SO_2$. Some people have carried out fruitful work on the synthesis methods and modification of catalysts and the regeneration of deactivated catalysts. Nevertheless, some aspects need to be further investigated, the resistance of cerium-based catalysts to different impurities at low temperature. Second, the traditional synthesis methods of catalysts also need further research to explore and develop new synthesis methods to enhance the interaction between active components and weaken the influence of impurities on the active site. Furthermore, in order to provide more excellent performance of cerium-based catalysts, it is necessary to further study the optimal ratio of active components. In addition, considering the cost of catalysts, some metal oxides have high costs, so the regeneration of poisoned catalysts is also a key research direction in the future.

**Author Contributions:** X.B., K.L.; literature search, writing-original draft preparation and editing, M.C., W.W., P.C.; writing-review and editing, X.B., W.W.; funding acquisition. All authors have read and agreed to the published version of the manuscript.

**Funding:** The work described above was supported by the Major State Basic Research Development Program of China (973 Program) [no. 2012CBA01205] and National Natural Science Foundation of China [no. 51274060].

**Data Availability Statement:** Not applicable.

**Conflicts of Interest:** The authors declare no conflict of interest.

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
