# Peer review of "Effects of Flue Gas Impurities on the Performance of Rare Earth Denitration Catalysts"

_catalysts, doi:10.3390/catal12080808_

Round 1
Reviewer 1 Report
Dear authors;
After a careful revision of your document entitled: "Effects of flue gas impurities on the performance of rare earth denitration catalysts", which provides an overall perspective of the topic related to usage of CeO2 materials to replace Vanadium content for selective catalytic reactions.
However major important issues should be addressed in order to improve the quality of the manuscript.
Revise as follows:
1) No comparison or data about the Vanadium catalyst is provided, thus, please add that information is relevant.
2) All images and graphs are extremely blurry, correct those. Also, provide more description in the image caption, that will allow a reader to understand more what is intend to communicate.
3) No information about high-resolution TEM or XRD is provided, remember that characterization takes an important role on understanding catalytic materials.
4) Data about DFT is vague and doesn´t provide much insights. Please check (https://doi.org/10.1016/j.cattod.2021.05.009) and use that information as part of your description for CeO2 catalyst.
5) Regarding English grammar a major improvements should be carried out. I took the liberty to mention a few on the PDF file, please revise my comments. However, I believe authors should seek for support from foreign language department. Many of the lines are not well written. Specially on the conclusion.

Author Response
Response to Reviewer 1 Comments
Thank you very much for your review and comments. We have carefully read the suggestions and comments and made some modifications to the manuscript. Here are the replies to your comments.
Point 1: No comparison or data about the Vanadium catalyst is provided, thus, please add that information is relevant.
Response 1: This issue has been supplemented
Figure 1(a) showed the activity comparisons of V-W/Ti and CeW catalysts and corresponding 1 wt % K-doped catalysts at temperature ranging from 100 to 300°C under a GHSV of 60 000 h-1. Without K doping, the activity of the CeW catalyst was slightly higher than the V-W/Ti catalyst below 280°C, with a maximum of nearly 99% NOx conversion at 220°C, maintained up to 300°C. During the NOx conversion test, when 1% K was loaded, the activity of the V-W/Ti catalyst dropped to 20% at 200°C, while the CeW catalyst remained above 70% at the same temperature. It can be concluded that the CeW catalyst is more resistant to alkali metals than the traditional vanadium-based catalyst V-W/Ti below 300°C. For CeW catalyst, The Na&K catalyst was less active at low temperatures but yielded higher NOx conversion above 200°C compared with the 1 K and 0.58 Na catalysts. At a given molar concentration, K gave rise to more deactivation than Na below 200°C, due to its more potent neutralizing properties.(Figure 1(b)).
In addition, Li et al. [56] also studied the effect of CaO on the V2O5–WO3/TiO2 and CeO2-WO3 catalyst. The results showed that CW catalyst had a better CaO resistance effect than VWTcatalyst for SCR(Figure 8).
Point 2: All images and graphs are extremely blurry, correct those. Also, provide more description in the image caption, that will allow a reader to understand more what is intend to communicate.
Response 2: This problem have been have been modified for the manuscript.
Point 3: No information about high-resolution TEM or XRD is provided, remember that characterization takes an important role on understanding catalytic materials.
Response 3: High-resolution TEM or XRD have been provided
Point 4: Data about DFT is vague and doesn’t provide much insights. Please check (https://doi.org/10.1016/j.cattod.2021.05.009) and use that information as part of your description for CeO2 catalyst.
Response 4: These good references play a vital in the integrity of our manuscript. Therefore, they have been cited on our manuscript.
Point 5: Regarding English grammar a major improvements should be carried out. I took the liberty to mention a few on the PDF file, please revise my comments. However, I believe authors should seek for support from foreign language department. Many of the lines are not well written. Specially on the conclusion.
Response 5: This problem have been have been modified for the manuscript.
Reviewer 2 Report
In this review, the authors have summarized the effects and deactivation mechanisms of various types of impurities on the activity of rare earth catalysts for NH3-SCR reaction. It has benefit for us to better development and application of NH3-SCR catalysts for rare earth denitration in the field of NOx control. After a minor revision for the current review, it maybe accepted. The details are shown as below.
1. 2.3 The influence of Pb. Pb is not an alkali or alkaline earth metal, it should be separated from the second part of “The influence of metal impurities”.
2. There are some literatures were not quoted, such as “Applied Catalysis B: Environmental, 2020, 270, 118860”. Please more carefully check the update reports in recent years.
3. The review should provide its own insights rather than simply summarize the results in the literature. Therefore, the authors should provide proper viewpoint in the end of each charpter.
4. It’s better to make a form to conclude the various types of impurities.
Author Response
Response to Reviewer 2 Comments
Thank you very much for your review and comments. We have carefully read the suggestions and comments and made some modifications to the manuscript. Here are the replies to your comments.
Point 1: 2.3 The influence of Pb. Pb is not an alkali or alkaline earth metal, it should be separated from the second part of “The influence of metal impurities”.
Response 1: This problem have been have been modified for the manuscript.
Point 2: There are some literatures were not quoted, such as “Applied Catalysis B: Environmental, 2020, 270, 118860”. Please more carefully check the update reports in recent years.
Response 2: I have checked the update reports in recent years.
These good references play a vital in the integrity of our manuscript. Therefore, they have been cited on our manuscript.
Point 3: The review should provide its own insights rather than simply summarize the results in the literature. Therefore, the authors should provide proper viewpoint in the end of each charpter.
Response 3: Modifications have been made.
In conclusion, although cerium-based catalysts have stronger resistance to metal impurities than vanadium-based catalysts, it still affects the SCR activity of catalysts. Therefore, how to improve the resistance of cerium-based catalysts to metal impurities has become the focus of future research. The catalyst can achieve better performance by adjusting the ratio of substances, different synthesis methods, or additives.
In short, non-metallic impurities widely exist in denitration flue gas, and the harm to the catalyst is inevitable. Rare earth catalysts with excellent performance and good resistance to impurities are bound to become the focus of industry research in the future, some people have achieved good results in this regard. Although it has excellent performance, some aspects need to be further explored. It is urgent to explore new synthesis methods and new material ratio.
Point 4: It’s better to make a form to conclude the various types of impurities.
Response 4: I have made a form to conclude the various types of impurities.
|
Types of impurities |
Effect on rare earth catalysts |
|
Na、K |
K and Na will decrease the acid sites of the catalyst, and their oxides and chlorides will weaken the reaction activity of the catalyst surface, inhibit the formation of oxygen vacancies and chemical adsorbed oxygen, thereby reducing the NH3 adsorption amount and weakening the denitrification performance. The influence of K is greater than that of Na, and the influence of oxides is more serious than that of chlorides. |
|
Ca |
Ca deposition can destroy the pore structure of the catalyst, reduce the surface active elements and acid sites, and Mg has a similar effect. |
|
Pb |
Lead will reduce the redox performance, chemical adsorption of oxygen and specific surface area of the catalyst. The toxicity of lead chloride is higher than lead oxide, because lead chloride is easier to form crystalline phase. |
|
P |
At low temperature, P promotes the grain growth of TiO2 and CeO2 in the catalyst, reduces the specific surface area of the catalyst, inhibits the electron transfer between Ce and Ti ions, and reduces its redox performance. At high temperature, P inhibits NOx and N2O produced by ammonia peroxidation, thereby increasing its activity. |
|
Cl |
HCl led to the decrease of specific surface area, the increase of crystallinity, the decrease of redox ability, and the significant decrease of surface acid sites, which further affected the catalyst activity. |
|
F |
F can inhibit crystallization, so that the catalyst has more surface active sites, increasing oxygen vacancies and chemisorption oxygen. In addition, the addition of F can bring more NOx adsorption sites and the formation of intermediate species, thereby promoting the activity of the catalyst. |
|
S |
Ce in rare earth catalysts can effectively delay the formation of surface sulfating substances, reduce the binding energy of ammonium and sulfate ions, so that ammonium sulfate is easier to decompose and improve the sulfur resistance of Ce modified catalysts. |
Round 2
Reviewer 1 Report
Dear Xue Bian;
After a careful inspection on your response to my revision comments on your manuscript entitled: Effects of flue gas impurities on the performance of rare earth denitration catalysts.
I´m honored to mention that your manuscript has been improved and all recommendations made fulfill.
I believe the manuscript is ready to be accepted, however, this is a duty and decision by the journal editor.
Great research, congrats!